# Image GANs meet Differentiable Rendering for Inverse Graphics and Interpretable 3D Neural Rendering

**Yuxuan Zhang**[1,4*]   **Wenzheng Chen**[1,2,3*]   **Huan Ling**[1,2,3]

**Jun Gao**[1,2,3]   **Yinan Zhang**[5]   **Antonio Torralba**[6]   **Sanja Fidler**[1,2,3]

NVIDIA[1]   University of Toronto[2]   Vector Institute[3]   University of Waterloo [4]
Stanford University[5]   MIT CSAIL[6]

{alezhang, wenzchen, huling, jung, sfidler}@nvidia.com, yinanzy@stanford.edu, torralba@mit.edu

## Abstract

Differentiable rendering has paved the way to training neural networks to perform "inverse graphics" tasks such as predicting 3D geometry from monocular photographs. To train high performing models, most of the current approaches rely on multi-view imagery which are not readily available in practice. Recent Generative Adversarial Networks (GANs) that synthesize images, in contrast, seem to acquire 3D knowledge implicitly during training: object viewpoints can be manipulated by simply manipulating the latent codes. However, these latent codes often lack further physical interpretation and thus GANs cannot easily be inverted to perform explicit 3D reasoning. In this paper, we aim to extract and disentangle 3D knowledge learned by generative models by utilizing differentiable renderers. Key to our approach is to exploit GANs as a multi-view data generator to train an inverse graphics network using an off-the-shelf differentiable renderer, and the trained inverse graphics network as a teacher to disentangle the GAN's latent code into interpretable 3D properties. The entire architecture is trained iteratively using cycle consistency losses. We show that our approach significantly outperforms state-of-the-art inverse graphics networks trained on existing datasets, both quantitatively and via user studies. We further showcase the disentangled GAN as a controllable 3D "neural renderer", complementing traditional graphics renderers.

## 1 Introduction

The ability to infer 3D properties such as geometry, texture, material, and light from photographs is key in many domains such as AR/VR, robotics, architecture, and computer vision. Interest in this problem has been explosive, particularly in the past few years, as evidenced by a large body of published works and several released 3D libraries (TensorflowGraphics by Valentin et al. (2019), Kaolin by J. et al. (2019), PyTorch3D by Ravi et al. (2020)).

The process of going from images to 3D is often called "inverse graphics", since the problem is inverse to the process of rendering in graphics in which a 3D scene is projected onto an image by taking into account the geometry and material properties of objects, and light sources present in the scene. Most work on inverse graphics assumes that 3D labels are available during training (Wang et al., 2018; Mescheder et al., 2019; Groueix et al., 2018; Wang et al., 2019; Choy et al., 2016), and trains a neural network to predict these labels. To ensure high quality 3D ground-truth, synthetic datasets such as ShapeNet (Chang et al., 2015) are typically used. However, models trained on synthetic datasets often struggle on real photographs due to the domain gap with synthetic imagery.

To circumvent these issues, recent work has explored an alternative way to train inverse graphics networks that sidesteps the need for 3D ground-truth during training. The main idea is to make

---

[*]indicates equal contribution.

Figure 1: We employ two "renderers": a GAN (StyleGAN in our work), and a differentiable graphics renderer (DIB-R in our work). We exploit StyleGAN as a synthetic data generator, and we label this data extremely efficiently. This "dataset" is used to train an inverse graphics network that predicts 3D properties from images. We use this network to disentangle StyleGAN's latent code through a carefully designed mapping network.

graphics renderers differentiable which allows one to infer 3D properties directly from images using gradient based optimization, Kato et al. (2018); Liu et al. (2019b); Li et al. (2018); Chen et al. (2019). These methods employ a neural network to predict geometry, texture and light from images, by minimizing the difference between the input image with the image rendered from these properties. While impressive results have been obtained in Liu et al. (2019b); Sitzmann et al. (2019); Liu et al. (2019a); Henderson & Ferrari (2018); Chen et al. (2019); Yao et al. (2018); Kanazawa et al. (2018), most of these works still require some form of implicit 3D supervision such as multi-view images of the same object with known cameras. Thus, most results have been reported on the synthetic ShapeNet dataset, or the large-scale CUB (Welinder et al., 2010) bird dataset annotated with keypoints from which cameras can be accurately computed using structure-from-motion techniques.

On the other hand, generative models of images appear to learn 3D information implicitly, where several works have shown that manipulating the latent code can produce images of the same scene from a different viewpoint (Karras et al., 2019a). However, the learned latent space typically lacks physical interpretation and is usually not disentangled, where properties such as the 3D shape and color of the object often cannot be manipulated independently.

In this paper, we aim to extract and disentangle 3D knowledge learned by generative models by utilizing differentiable graphics renderers. We exploit a GAN, specifically StyleGAN (Karras et al., 2019a), as a generator of multi-view imagery to train an inverse graphics neural network using a differentiable renderer. In turn, we use the inverse graphics network to inform StyleGAN about the image formation process through the knowledge from graphics, effectively disentangling the GAN's latent space. We connect StyleGAN and the inverse graphics network into a single architecture which we iteratively train using cycle-consistency losses. We demonstrate our approach to significantly outperform inverse graphics networks on existing datasets, and showcase controllable 3D generation and manipulation of imagery using the disentangled generative model.

## 2 RELATED WORK

**3D from 2D:** Reconstructing 3D objects from 2D images is one of the mainstream problems in 3D computer vision. We here focus our review to single-image 3D reconstruction which is the domain of our work. Most of the existing approaches train neural networks to predict 3D shapes from images by utilizing 3D labels during training, Wang et al. (2018); Mescheder et al. (2019); Choy et al. (2016); Park et al. (2019). However, the need for 3D training data limits these methods to the use of synthetic datasets. When tested on real imagery there is a noticeable performance gap.

Newer works propose to differentiate through the traditional rendering process in the training loop of neural networks, Loper & Black (2014); Kato et al. (2018); Liu et al. (2019b); Chen et al. (2019); Petersen et al. (2019); Gao et al. (2020). Differentiable renderers allow one to infer 3D from 2D images without requiring 3D ground-truth. However, in order to make these methods work in practice, several additional losses are utilized in learning, such as the multi-view consistency loss whereby the cameras are assumed known. Impressive reconstruction results have been obtained on the synthetic ShapeNet dataset. While CMR by Kanazawa et al. (2018) and DIB-R by Chen et al. (2019) show real-image 3D reconstructions on CUB and Pascal3D (Xiang et al., 2014) datasets, they rely on manually annotated keypoints, while still failing to produce accurate results.

A few recent works, Wu et al. (2020); Li et al. (2020); Goel et al. (2020); Kato & Harada (2019), explore 3D reconstruction from 2D images in a completely unsupervised fashion. They recover both 3D shapes and camera viewpoints from 2D images by minimizing the difference between original and re-projected images with additional unsupervised constraints, e.g., semantic information (Li et al. (2020)), symmetry (Wu et al. (2020)), GAN loss (Kato & Harada (2019)) or viewpoint distribution (Goel et al. (2020)). Their reconstruction is typically limited to 2.5D (Wu et al. (2020)),

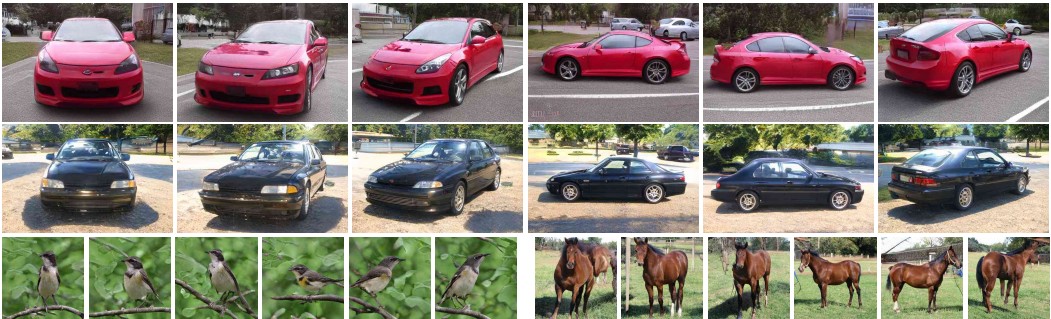

Figure 2: We show examples of cars (first two rows) synthesized in chosen viewpoints (columns). To get these, we fix the latent code $w_v^*$ that controls the viewpoint (one code per column) and randomly sample the remaining dimensions of (Style)GAN's latent code (to get rows). Notice how well aligned the two cars are in each column. In the third row we show the same approach applied to horse and bird StyleGAN.

and produces lower quality results than when additional supervision is used (Goel et al. (2020); Li et al. (2020); Kato & Harada (2019)). In contrast, we utilize GANs to generate multi-view realistic datasets that can be annotated *extremely efficiently*, which leads to accurate 3D results. Furthermore, our model achieves disentanglement in GANs and turns them into interpretable 3D neural renderers.

**Neural Rendering with GANs:** GANs (Goodfellow et al., 2014; Karras et al., 2019a) can be regarded as neural renderers, as they take a latent code as input and "render" an image. However, the latent code is sampled from a predefined prior and lacks interpretability. Several works generate images with conditions: a semantic mask (Zhu et al., 2017), scene layout Karacan et al. (2016), or a caption (Reed et al., 2016), and manipulate the generated images by modifying the input condition. Despite tremendous progress in this direction, there is little work on generating images through an interpretable 3D physics process. Dosovitskiy et al. (2016) synthesizes images conditioned on object style, viewpoint, and color. Most relevant work to ours is Zhu et al. (2018), which utilizes a learnt 3D geometry prior and generates images with a given viewpoint and texture code. We differ in three important ways. First, we do not require a 3D dataset to train the 3D prior. Second, the texture in our model has 3D physical meaning, while Zhu et al. (2018) still samples from a prior. We further control background while Zhu et al. (2018) synthesizes objects onto white background.

**Disentangling GANs:** Learning disentangled representations has been widely explored, Lee et al. (2020); Lin et al. (2019); Perarnau et al. (2016). Representative work is InfoGAN Chen et al. (2016), which tries to maximize the mutual information between the prior and the generated image distribution. However, the disentangled code often still lacks physical interpretability. Tewari et al. (2020) transfers face rigging information from an existing model to control face attribute disentanglement in the StyleGAN latent space. Shen et al. (2020) aims to find the latent space vectors that correspond to meaningful edits, while Härkönen et al. (2020) exploits PCA to disentangle the latent space. Parallel to our work, Zhang et al. (2021); Li et al. (2021) attempt to interpret the semantic meaning of StyleGAN latent space. In our work, we disentangle the latent space with knowledge from graphics.

## 3 OUR APPROACH

We start by providing an overview of our approach (Fig. 1), and describe the individual components in more detail in the following sections. Our approach marries two types of renderers: a GAN-based neural "renderer" and a differentiable graphics renderer. Specifically, we leverage the fact that the recent state-of-the-art GAN architecture StyleGAN by Karras et al. (2019a;b) learns to produce highly realistic images of objects, and allows for a reliable control over the camera. We manually select a few camera views with a rough viewpoint annotation, and use StyleGAN to generate a large number of examples per view, which we explain in Sec. 3.1. In Sec. 3.2, we exploit this dataset to train an inverse graphics network utilizing the state-of-the-art differentiable renderer, DIB-R by Chen et al. (2019) in our work, with a small modification that allows it to deal with noisy cameras during training. In Sec. 3.3, we employ the trained inverse graphics network to disentangle StyleGAN's latent code and turn StyleGAN into a 3D neural renderer, allowing for control over explicit 3D properties. We fine-tune the entire architecture, leading to significantly improved results.

### 3.1 STYLEGAN AS SYNTHETIC DATA GENERATOR

We first aim to utilize StyleGAN to generate multi-view imagery. StyleGAN is a 16 layers neural network that maps a latent code $z \in Z$ drawn from a normal distribution into a realistic image. The code $z$ is first mapped to an intermediate latent code $w \in W$ which is transformed to $w^* =$

$(w_1^*, w_2^*, ..., w_{16}^*) \in W^*$ through 16 learned affine transformations. We call $W^*$ the transformed latent space to differentiate it from the intermediate latent space $W$. Transformed latent codes $w^*$ are then injected as the style information to the StyleGAN Synthesis network.

Different layers control different image attributes. As observed in Karras et al. (2019a), styles in early layers adjust the camera viewpoint while styles in the intermediate and higher layers influence shape, texture and background. We provide a careful analysis of all layers in Appendix. We empirically find that the latent code $w_v^* := (w_1^*, w_2^*, w_3^*, w_4^*)$ in the first 4 layers controls camera viewpoints. That is, if we sample a new code $w_v^*$ but keep the remaining dimensions of $w^*$ fixed (which we call the conten code), we generate images of the same object depicted in a different viewpoint. Examples are shown in Fig. 2.

We further observe that a sampled code $w_v^*$ in fact represents a fixed camera viewpoint. That is, if we keep $w_v^*$ fixed but sample the remaining dimensions of $w^*$, StyleGAN produces imagery of different objects in the same camera viewpoint. This is shown in columns in Fig. 2. Notice how aligned the objects are in each of the viewpoints. This makes StyleGAN a *multi-view* data generator!

**"StyleGAN" multi-view dataset:** We manually select several views, which cover all the common viewpoints of an object ranging from 0-360 in azimuth and roughly 0-30 in elevation. We pay attention to choosing viewpoints in which the objects look most consistent. Since inverse graphics works require camera pose information, we annotate the chosen viewpoint codes with a rough absolute camera pose. To be specific, we classify each viewpoint code into one of 12 azimuth angles, uniformly sampled along $360 \deg$. We assign each code a fixed elevation ($0°$) and camera distance. These camera poses provide a very coarse annotation of the actual pose – the annotation serves as the initialization of the camera which we will optimize during training. This allows us to annotate all views (and thus the entire dataset) in **only 1 minute** – making annotation effort neglible. For each viewpoint, we sample a large number of content codes to synthesize different objects in these views. Fig. 2 shows 2 cars, and a horse and a bird. Appendix provides more examples.

Since DIB-R also utilizes segmentation masks during training, we further apply MaskRCNN by He et al. (2017) to get instance segmentation in our generated dataset. As StyleGAN sometimes generates unrealistic images or images with multiple objects, we filter out "bad" images which have more than one instance, or small masks (less than 10% of the whole image area).

## 3.2 Training an Inverse Graphics Neural Network

Following CMR by Kanazawa et al. (2018), and DIB-R by Chen et al. (2019), we aim to train a 3D prediction network $f$, parameterized by $\theta$, to infer 3D shapes (represented as meshes) along with textures from images. Let $I_V$ denote an image in viewpoint $V$ from our StyleGAN dataset, and $M$ its corresponding object mask. The inverse graphics network makes a prediction as follows: $\{S, T\} = f_\theta(I_V)$, where $S$ denotes the predicted shape, and $T$ a texture map. Shape $S$ is deformed from a sphere as in Chen et al. (2019). While DIB-R also supports prediction of lighting, we empirically found its performance is weak for realistic imagery and we thus omit lighting estimation in our work.

To train the network, we adopt DIB-R as the differentiable graphics renderer that takes $\{S, T\}$ and $V$ as input and produces a rendered image $I_V' = r(S, T, V)$ along with a rendered mask $M'$. Following DIB-R, the loss function then takes the following form:

$$L(I, S, T, V; \theta) = \lambda_{\text{col}} L_{\text{col}}(I, I') + \lambda_{\text{percpt}} L_{\text{pecept}}(I, I') + L_{\text{IOU}}(M, M')$$
$$+ \lambda_{\text{sm}} L_{\text{sm}}(S) + \lambda_{\text{lap}} L_{\text{lap}}(S) + \lambda_{\text{mov}} L_{\text{mov}}(S) \tag{1}$$

Here, $L_{\text{col}}$ is the standard $L_1$ image reconstruction loss defined in the RGB color space while $L_{\text{percpt}}$ is the perceptual loss that helps the predicted texture look more realistic. Note that rendered images do not have background, so $L_{\text{col}}$ and $L_{\text{percept}}$ are calculated by utilizing the mask. $L_{\text{IOU}}$ computes the intersection-over-union between the ground-truth mask and the rendered mask. Regularization losses such as the Laplacian loss $L_{\text{lap}}$ and flatten loss $L_{\text{sm}}$ are commonly used to ensure that the shape is well behaved. Finally, $L_{\text{mov}}$ regularizes the shape deformation to be uniform and small.

Since we also have access to multi-view images for each object we also include a multi-view consistency loss. In particular, our loss per object $k$ is:

$$\mathcal{L}_k(\theta) = \sum_{i,j, i \neq j} \left( L(I_{V_i^k}, S_k, T_k, V_i^k; \theta) + L(I_{V_j^k}, S_k, T_k, V_j^k; \theta) \right) \tag{2}$$

$$\text{where } \{S_k, T_k, L_k\} = f_\theta(I_{V_i^k})$$

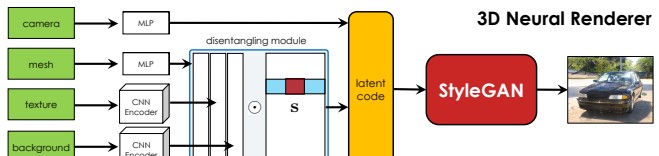

Figure 3: A mapping network maps camera, shape, texture and background into a disentangled code that is passed to StyleGAN for "rendering". We refer to this network as StyleGAN-R.

While more views provide more constraints, empirically, two views have been proven sufficient. We randomly sample view pairs $(i, j)$ for efficiency.

We use the above loss functions to jointly train the neural network $f$ and optimize viewpoint cameras $V$ (which were fixed in Chen et al. (2019)). We assume that different images generated from the same $w_v^*$ correspond to the same viewpoint $V$. Optimizing the camera jointly with the weights of the network allows us to effectively deal with noisy initial camera annotations.

## 3.3 DISENTANGLING STYLEGAN WITH THE INVERSE GRAPHICS MODEL

The inverse graphics model allows us to infer a 3D mesh and texture from a given image. We now utilize these 3D properties to disentangle StyleGAN's latent space, and turn StyleGAN into a fully controllable 3D neural renderer, which we refer to as StyleGAN-R. Note that StyleGAN in fact synthesizes more than just an object, it also produces the background, i.e., the entire scene. Ideally we want control over the background as well, allowing the neural renderer to render 3D objects into desired scenes. To get the background from a given image, we simply mask out the object.

We propose to learn a mapping network to map the viewpoint, shape (mesh), texture and background into the StyleGAN's latent code. Since StyleGAN may not be completely disentangled, we further fine-tune the entire StyleGAN model while keeping the inverse graphics network fixed.

**Mapping Network:** Our mapping network, visualized in Figure 3, maps the viewpoints to first 4 layers and maps the shape, texture and background to the last 12 layers of $W^*$. For simplicity, we denote the first 4 layers as $W_V^*$ and the last 12 layers as $W_{STB}^*$, where $W_V^* \in \mathbb{R}^{2048}$ and $W_{STB}^* \in \mathbb{R}^{3008}$. Specifically, the mapping network $g_v$ for viewpoint $V$ and $g_s$ for shape $S$ are separate MLPs while $g_t$ for texture $T$ and $g_b$ for background $B$ are CNN layers:

$$\mathbf{z}^{\text{view}} = g_v(V; \theta_v), \ \mathbf{z}^{\text{shape}} = g_s(S; \theta_s), \mathbf{z}^{\text{txt}} = g_t(T; \theta_t), \ \mathbf{z}^{\text{bck}} = g_b(B; \theta_b), \tag{3}$$

where $\mathbf{z}^{\text{view}} \in \mathbb{R}^{2048}, \mathbf{z}^{\text{shape}}, \mathbf{z}^{\text{txt}}, \mathbf{z}^{\text{bck}} \in \mathbb{R}^{3008}$ and $\theta_v, \theta_s, \theta_t, \theta_b$ are network parameters. We softly combine the shape, texture and background codes into the final latent code as follows:

$$\tilde{w}^{mtb} \quad = \quad \mathbf{s}^{\text{m}} \odot \mathbf{z}^{\text{shape}} + \mathbf{s}^{\text{t}} \odot \mathbf{z}^{\text{txt}} + \mathbf{s}^{\text{b}} \odot \mathbf{z}^{\text{bck}}, \tag{4}$$

where $\odot$ denotes element-wise product, and $\mathbf{s}^{\text{m}}, \mathbf{s}^{\text{t}}, \mathbf{s}^{\text{b}} \in \mathbb{R}^{3008}$ are shared across all the samples. To achieve disentanglement, we want each dimension of the final code to be explained by only one property (shape, texture or background). We thus normalize each dimension of $\mathbf{s}$ using softmax.

In practice, we found that mapping $V$ to a high dimensional code is challenging since our dataset only contains a limited number of views, and $V$ is limited to azimuth, elevation and scale. We thus map $V$ to the subset of $W_V^*$, where we empirically choose 144 of the 2048 dimensions with the highest correlation with the annotated viewpoints. Thus, $\mathbf{z}^{\text{view}} \in \mathbb{R}^{144}$ in our case.

**Training Scheme:** We train the mapping network and fine-tune StyleGAN in two separate stages. We first freeze StyleGAN's weights and train the mapping network only. This warms up the mapping network to output reasonable latent codes for StyleGAN. We then fine-tune both StyleGAN and the mapping network to better disentangle different attributes. We provide details next.

In the warm up stage, we sample viewpoint codes $w_v^*$ among the chosen viewpoints, and sample the remaining dimensions of $w^* \in W^*$. We try to minimize the $L_2$ difference between the mapped code $\tilde{w}$ and StyleGAN's code $w^*$. To encourage the disentanglement in the latent space, we penalize the entropy of each dimension $i$ of $\mathbf{s}$. Our overall loss function for our mapping network is:

$$L_{\text{mapnet}}(\theta_v, \theta_s, \theta_t, \theta_v) = ||\tilde{w} - w^*||_2 - \sum_i \sum_{k \in \{m,t,b\}} \mathbf{s}_i^k \log(\mathbf{s}_i^k). \tag{5}$$

By training the mapping network, we find that view, shape and texture can be disentangled in the original StyleGAN model but the background remains entangled. We thus fine-tune the model to get a better disentanglement. To fine-tune the StyleGAN network we incorporate a cycle consistency loss. In particular, by feeding a sampled shape, texture and background to StyleGAN we

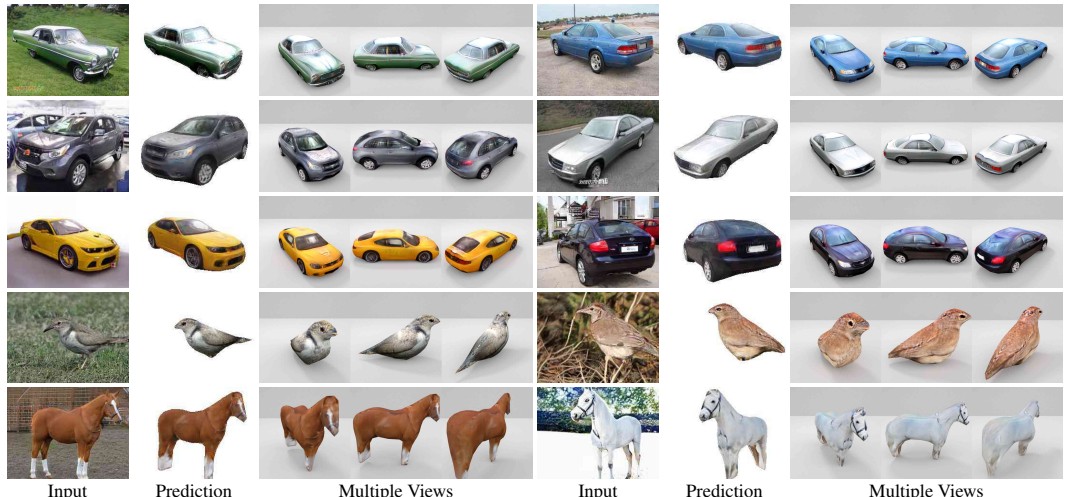

Figure 4: **3D Reconstruction Results:** Given input images (1st column), we predict 3D shape, texture, and render them into the same viewpoint (2nd column). We also show renderings in 3 other views in remaining columns to showcase 3D quality. Our model is able to reconstruct cars with various shapes, textures and viewpoints. We also show the same approach on harder (articulated) objects, i.e., bird and horse.

obtain a synthesized image. We encourage consistency between the original sampled properties and the shape, texture and background predicted from the StyleGAN-synthesized image via the inverse graphics network. We further feed the same background $B$ with two different $\{S, T\}$ pairs to generate two images $I_1$ and $I_2$. We then encourage the re-synthesized backgrounds $\bar{B}_1$ and $\bar{B}_2$ to be similar. This loss tries to disentangle the background from the foreground object. During training, we find that imposing the consistency loss on $B$ in image space results in blurry images, thus we constrain it in the code space. Our fine-tuning loss takes the following form:

$$L_{\text{stylegan}}(\theta_{\text{gan}}) = ||S - \bar{S}||_2 + ||T - \bar{T}||_2 + ||g_b(B) - g_b(\bar{B})||_2 + ||g_b(\bar{B}_1) - g_b(\bar{B}_2)||_2 \quad (6)$$

## 4 EXPERIMENTS

In this section, we showcase our approach on inverse graphics tasks (3D image reconstruction), as well as on the task of 3D neural rendering and 3D image manipulation.

**Image Datasets for training StyleGAN:** We use three category-specific StyleGAN models, one representing a rigid object class, and two representing articulated (and thus more challenging) classes. We use the official car and horse model from StyleGAN2 (Karras et al., 2019b) repo which are trained on LSUN Car and LSUN Horse with 5.7M and 2M images. We also train a bird model on NABirds (Van Horn et al., 2015) dataset, which contains 48k images.

**Our "StyleGAN" Dataset:** We first randomly sample 6000 cars, 1000 horse and 1000 birds with diverse shapes, textures, and backgrounds from StyleGAN. After filtering out images with bad masks as described in Sec. 3, 55429 cars, 16392 horses and 7948 birds images remain in our dataset which is significant larger than the Pascal3D car dataset (Xiang et al., 2014) (4175 car images). Note that nothing prevents us from synthesizing a significantly larger amount of data, but in practice, this amount turned out to be sufficient to train good models. We provide more examples in Appendix.

### 4.1 3D RECONSTRUCTION RESULTS

**Training Details:** Our DIB-R based inverse graphics model was trained with Adam (Kingma & Ba (2015)), with a learning rate of 1e-4. We set $\lambda_{\text{IOU}}$, $\lambda_{\text{col}}$, $\lambda_{\text{lap}}$, $\lambda_{\text{sm}}$ and $\lambda_{\text{mov}}$ to 3, 20, 5, 5, and 2.5, respectively. We first train the model with $L_{\text{col}}$ loss for 3K iterations, and then fine-tune the model by adding $L_{\text{pecept}}$ to make the texture more realistic. We set $\lambda_{\text{percept}}$ to 0.5. The model converges in 200K iterations with batch size 16. Training takes around 120 hours on four V100 GPUs.

**Results:** We show 3D reconstruction results in Fig. 4. Notice the quality of the predicted shapes and textures, and the diversity of the 3D car shapes we obtain. Our method also works well on more challenging (articulated) classes, e.g. horse and bird. We provide additional examples in Appendix.

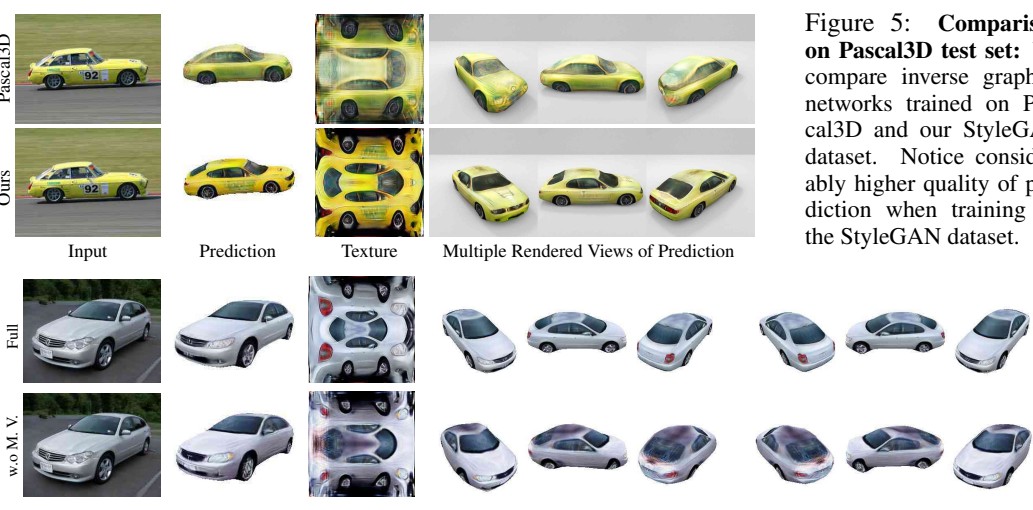

Figure 5: **Comparison on Pascal3D test set:** We compare inverse graphics networks trained on Pascal3D and our StyleGAN dataset. Notice considerably higher quality of prediction when training on the StyleGAN dataset.

Figure 6: **Ablation Study:** We ablate the use of multi-view consistency loss. Both texture are shape are worse without this loss, especially in the invisible parts (rows 2, 5, denoted by "w.o M. V." – no multi-view consistency used during training), showcasing the importance of our StyleGAN-multivew dataset.

**Qualitative Comparison:** To showcase our approach, we compare our inverse graphics network trained on our StyleGAN dataset with exactly the same model but which we train on the Pascal3D car dataset. Pascal3D dataset has annotated keypoints, which we utilize to train the baseline model, termed as as Pascal3D-model. We show qualitative comparison on Pascal3D test set in Fig. 5. Note that the images from Pascal3D dataset are different from those our StyleGAN-model was trained on. Although the Pascal3D-model's prediction is visually good in the input image view, rendered predictions in other views are of noticeably lower quality than ours, which demonstrates that we recover 3D geometry and texture better than the baseline.

| Dataset | Size | Annotation |
|---|---|---|
| Pascal3D | 4K | 200-350h |
| StyleGAN | **50K** | **~1min** |

(a) Dataset Comparison

| Model | Pascal3D test | StyleGAN test |
|---|---|---|
| Pascal3D | **0.80** | 0.81 |
| Ours | 0.76 | **0.95** |

(b) 2D IOU Evaluation

| | Overall | Shape | Texture |
|---|---|---|---|
| Ours | **57.5%** | **61.6%** | **56.3%** |
| Pascal3D-model | 25.9% | 26.4% | 32.8% |
| No Preference | 16.6% | 11.9% | 10.8% |

(c) User Study

Table 1: **(a):** We compare dataset size and annotation time of Pascal3D with our StyleGAN dataset. **(b):** We evaluate re-projected 2D IOU score of our StyleGAN-model vs the baseline Pascal3D-model on the two datasets. **(c):** We conduct a user study to judge the quality of 3D estimation.

**Quantitative Comparison:** We evaluate the two networks in Table 1 for the car class. We report the estimated annotation time in Table. 1 (a) to showcase efficiency behind our StyleGAN dataset. It takes 3-5 minutes to annotate keypoints for one object, which we empirically verify. Thus, labeling Pascal3D required around 200-350 hours while ours takes only 1 minute to annotate a 10 times larger dataset. In Table 1 (b), we evaluate shape prediction quality by the re-projected 2D IOU score. Our model outperforms the Pascal3D-model on the SyleGAN test set while Pascal3D-model is better on the Pascal test set. This is not surprising since there is a domain gap between two datasets and thus each one performs best on their own test set. Note that this metric only evaluates quality of the prediction in input view and thus not reflect the actual quality of the predicted 3D shape/texture.

To analyze the quality of 3D prediction, we conduct an AMT user study on the *Pascal3D test set* which contains 220 images. We provide users with the input image and predictions rendered in 6 views (shown in Fig. 5, right) for both models. We ask them to choose the model with a more realistic shape and texture prediction that matches the input object. We provide details of the study in the Appendix. We report results in Table. 1 (c). Users show significant preference of our results versus the baseline, which confirms that the quality of our 3D estimation.

**Ablation study:** In Fig 6 we ablate the importance of using multiple views in our dataset, i.e., by encouraging multi-view consistency loss during training. We compare predictions from inverse graphics networks trained with and without this loss, with significant differences in quality.

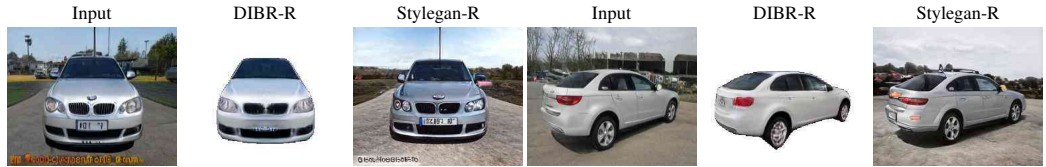

Figure 7: **Dual Renderer:** Given input images (1st column), we first predict mesh and texture, and render them with the graphics renderer (2nd column), and our StyleGAN-R (3rd column).

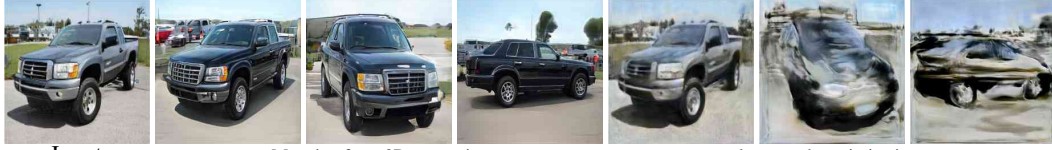

Figure 8: **Latent code manipulation:** Given an input image (col 1), we predict 3D properties and synthesize a new image with StyleGAN-R, by manipulating the viewpoint (col 2, 3, 4). Alternatively, we directly optimize the (original) StyleGAN latent code w.r.t. image, however this leads to a blurry reconstruction (col 5). Moreover, when we try to adjust the style for the optimized code, we get low quality results (col 6, 7).

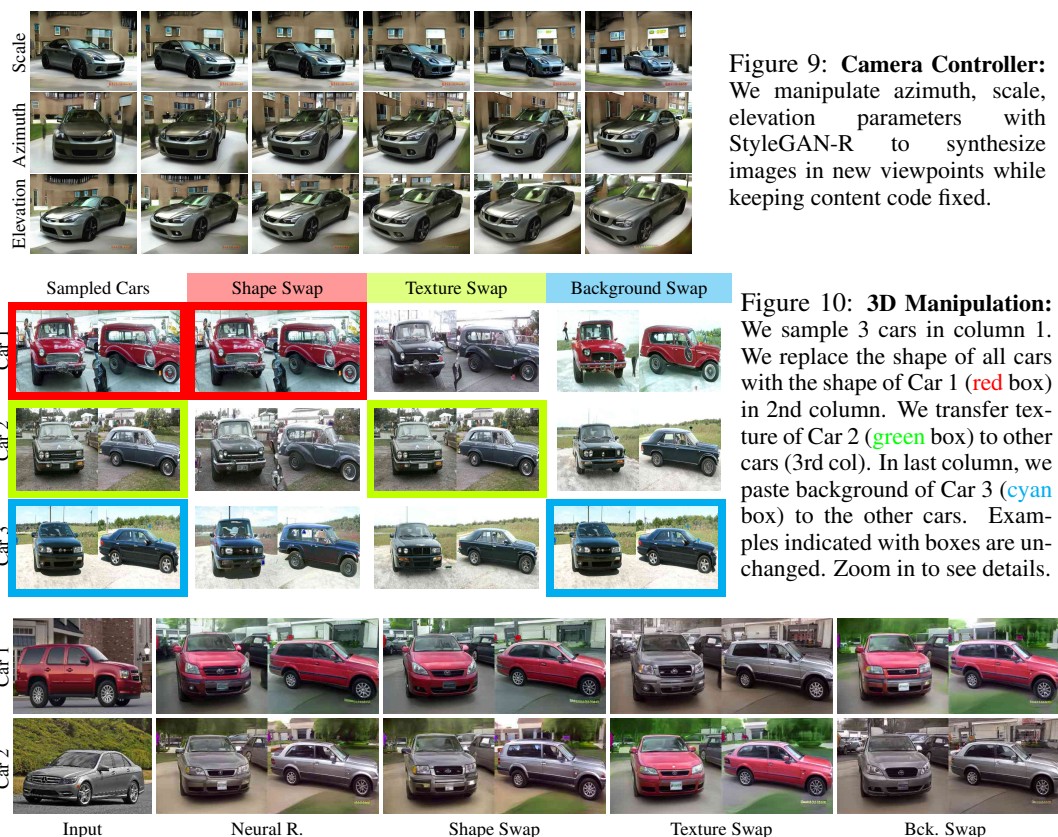

Figure 9: **Camera Controller:** We manipulate azimuth, scale, elevation parameters with StyleGAN-R to synthesize images in new viewpoints while keeping content code fixed.

Figure 10: **3D Manipulation:** We sample 3 cars in column 1. We replace the shape of all cars with the shape of Car 1 (red box) in 2nd column. We transfer texture of Car 2 (green box) to other cars (3rd col). In last column, we paste background of Car 3 (cyan box) to the other cars. Examples indicated with boxes are unchanged. Zoom in to see details.

Figure 11: **Real Image Manipulation:** Given input images (1st col), we predict 3D properties and use our StyleGAN-R to render them back (2nd col). We swap out shape, texture & background in cols 3-5.

## 4.2 DUAL RENDERERS

**Training Details:** We train StyleGAN-R using Adam with learning rate of 1e-5 and batch size 16. Warmup stage takes 700 iterations, and we perform joint fine-tuning for another 2500 iterations.

With the provided input image, we first predict mesh and texture using the trained inverse graphics model, and then feed these 3D properties into StyleGAN-R to generate a new image. For comparison, we feed the same 3D properties to the DIB-R graphics renderer (which is the OpenGL renderer). Results are provided in Fig. 7. Note that DIB-R can only render the predicted object, while StyleGAN-R also has the ability to render the object into a desired background. We find that StyleGAN-R produces relatively consistent images compared to the input image. Shape and texture are well preserved, while only the background has a slight content shift.

### 4.3 3D Image Manipulation with StyleGAN-R

We test our approach in manipulating StyleGAN-synthesized images from our test set and real images. Specifically, given an input image, we predict 3D properties using the inverse graphics network, and extract background by masking out the object with Mask-RCNN. We then manipulate and feed these properties to StyleGAN-R to synthesize new views.

**Controlling Viewpoints:** We first freeze shape, texture and background, and change the camera viewpoint. Example is shown in Fig. 9. We obtain meaningful results, particularly for shape and texture. For comparison, an alternative way that has been explored in literature is to directly optimize the GAN's latent code (in our case the original StyleGAN's code) via an L2 image reconstruction loss. Results are shown in the last three columns in Fig. 8. As also observed in Abdal et al. (2019), this approach fails to generate plausible images, showcasing the importance of the mapping network and fine-tuning the entire architecture with 3D inverse graphics network in the loop.

**Controlling Shape, Texture and Background:** We further aim to manipulate 3D properties, while keeping the camera viewpoint fixed. In the second column of Fig 10, we replace the shapes of all cars to one chosen shape (red box) and perform neural rendering using StyleGAN-R. We successfully swap the shape of the car while maintaining other properties. We are able to modify tiny parts of the car, such as trunk and headlights. We do the same experiment but swapping texture and background in the third and forth column of Fig 10. We notice that swapping textures also slightly modifies the background, pointing that further improvements are possible in disentangling the two.

**Real Image Editing:** As shown in Fig. 11, our framework also works well when provided with real images, since StyleGAN's images, which we use in training, are quite realistic.

### 4.4 Limitations

While recovering faithful 3D gemetry and texture, our model fails to predict correct lighting. Real images and StyleGAN-generated images contain advanced lighting effects such as reflection, transparency and shadows, and our spherical harmonic lighting model is incapable in dealing with it successfully. We also only partly succeed at disentangling the background, which one can see by noticing slight changes in background in Fig. 7, Fig. 10 and Fig. 11. Predicting faithful shapes for out-of-distribution objects as discussed in Appendix is also a significant challenge. We leave improvements to future work.

## 5 Conclusion

In this paper, we introduced a new powerful architecture that links two renderers: a state-of-the-art image synthesis network and a differentiable graphics renderer. The image synthesis network generates training data for an inverse graphics network. In turn, the inverse graphics network teaches the synthesis network about the physical 3D controls. We showcased our approach to obtain significantly higher quality 3D reconstruction results while requiring $10,000\times$ less annotation effort than standard datasets. We also provided 3D neural rendering and image manipulation results demonstrating the effectiveness of our approach.

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

# APPENDIX

## A  OVERVIEW

In the Appendix, we first show feature visualization of StyleGAN layers in Sec. B. We then provide a detailed explanation of our StyleGAN dataset creation in Sec. C, including examples of the generated images and selected viewpoints. Next, we do a systematic analysis of our camera initialization method in Sec. D. Finally, we show additional results on the 3D inverse graphics task in Sec. E, additional details of the user study in Sec. F, futher examples of StyleGAN disentanglement in Sec. G, with ablation studies and a discussion of limitations in Sec. H and Sec. K, respectively.

## B  STYLEGAN LAYERS VISUALIZATION

The official StyleGAN code repository provides models of different object categories at different resolutions. Here we take the $512 \times 384$ car model as the example. This model contains 16 layers, where every two consecutive layers form a block. Each block has a different number of channels. In the last block, the model produces a 32-channel feature map at a $512 \times 384$ resolution. Finally, a learned RGB transformation function is applied to convert the feature map into an RGB image.

We visualize the feature map for each block via the learned RGB transformation function. Specifically, for the feature map in each block with the size of $h \times w \times c$, we first sum along the feature dimension, forming a $h \times w \times 1$ tensor. We then repeat the feature 32 times and generate a $h \times w \times 32$ new feature map. This allows us to keep the information of all the channels and directly apply the RGB transformation function in the last block to convert it to the RGB image.

As shown in Fig A, we find that blocks 1 and 2 do not exhibit interpretable structure while the car shape starts to appear in blocks 3-5. We observe that there is a rough car contour in block 4 which further becomes clear in block 5. From blocks 6 to 8, the car's shape becomes increasingly finer and background scene also appears. This supports some of our findings, i.e., the viewpoint is controlled in block 1 and 2 (first 4 layers) while shape, texture, and background exist in the last 12 layers.

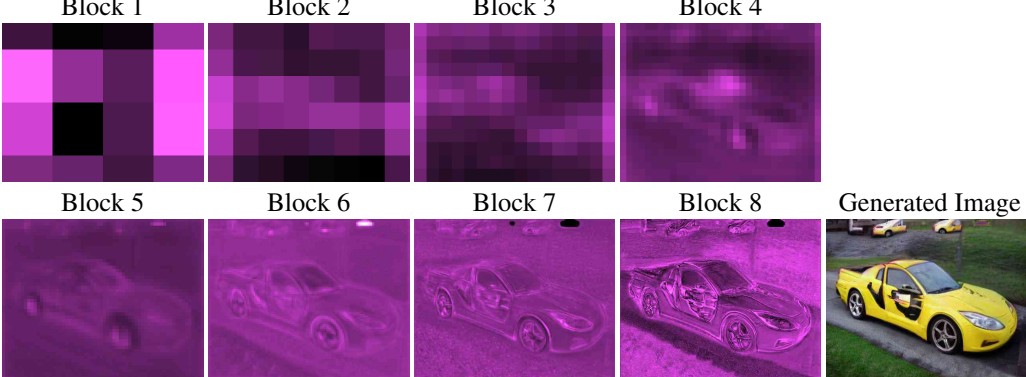

Figure A: **Layer Visualization for Each Block**: Notice that the car contour starts to appear in blocks 4 and higher. This supports some of our findings that the early blocks control viewpoint (and other global properties), while shape, texture and background are controlled in the higher layers.

## C  OUR "STYLEGAN" DATASET

We visualize all of our selected viewpoints in our dataset in Fig. B. Our car training dataset contains 39 viewpoints. For the horse and bird datasets, we choose 22 and 8 views, respectively. We find that these views are sufficient to learn accurate 3D inverse graphics networks. We could not find views that would depict the object from a higher up camera, i.e., a viewpoint from which the roof of the car or the back of the horse would be more clearly visible. This is mainly due to the original dataset on which StyleGAN was trained on, which lacked such views. This leads to challenges in training inverse graphics networks to accurately predict the top of the objects.

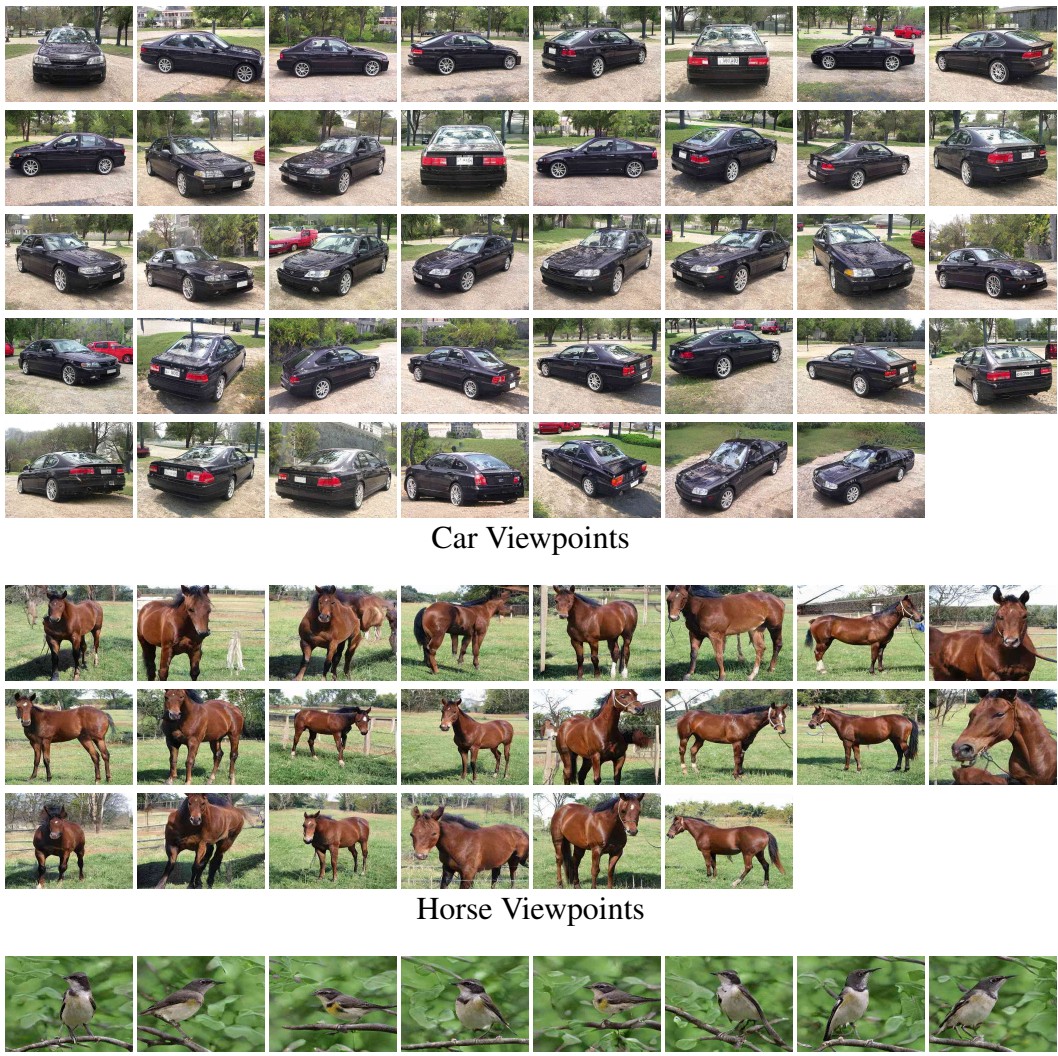

Car Viewpoints

Horse Viewpoints

Bird Viewpoints

Figure B: **All Viewpoints:** We show an example of a car, bird and a horse synthesized in all of our chosen viewpoints. While shape and texture are not perfectly consistent across views, they are sufficiently accurate to enable training accurate inverse graphics networks in our downstream tasks. Horses and birds are especially challenging due to articulation. One can notice small changes in articulation across viewpoints. Dealing with articulated objects is subject to future work.

Notice the high consistency of both the car shape and texture as well as the background scene across the different viewpoints. Note that for articulated objects such as the horse and bird classes, StyleGAN does not perfectly preserve object articulation in different viewpoints, which leads to challenges in training high accuracy models using multi-view consistency loss. We leave further investigation of articulated objects to future work.

We further show examples from our StyleGAN-generated dataset in Fig. C. Our dataset contains objects with various shapes, textures and viewpoints. In particular, in the first six rows, one can notice a diverse variants of car types (Standard Car, SUV, Sports car, Antique Car, etc) . We find that StyleGAN can also produce rare car shapes like trucks, but with a lower probability.

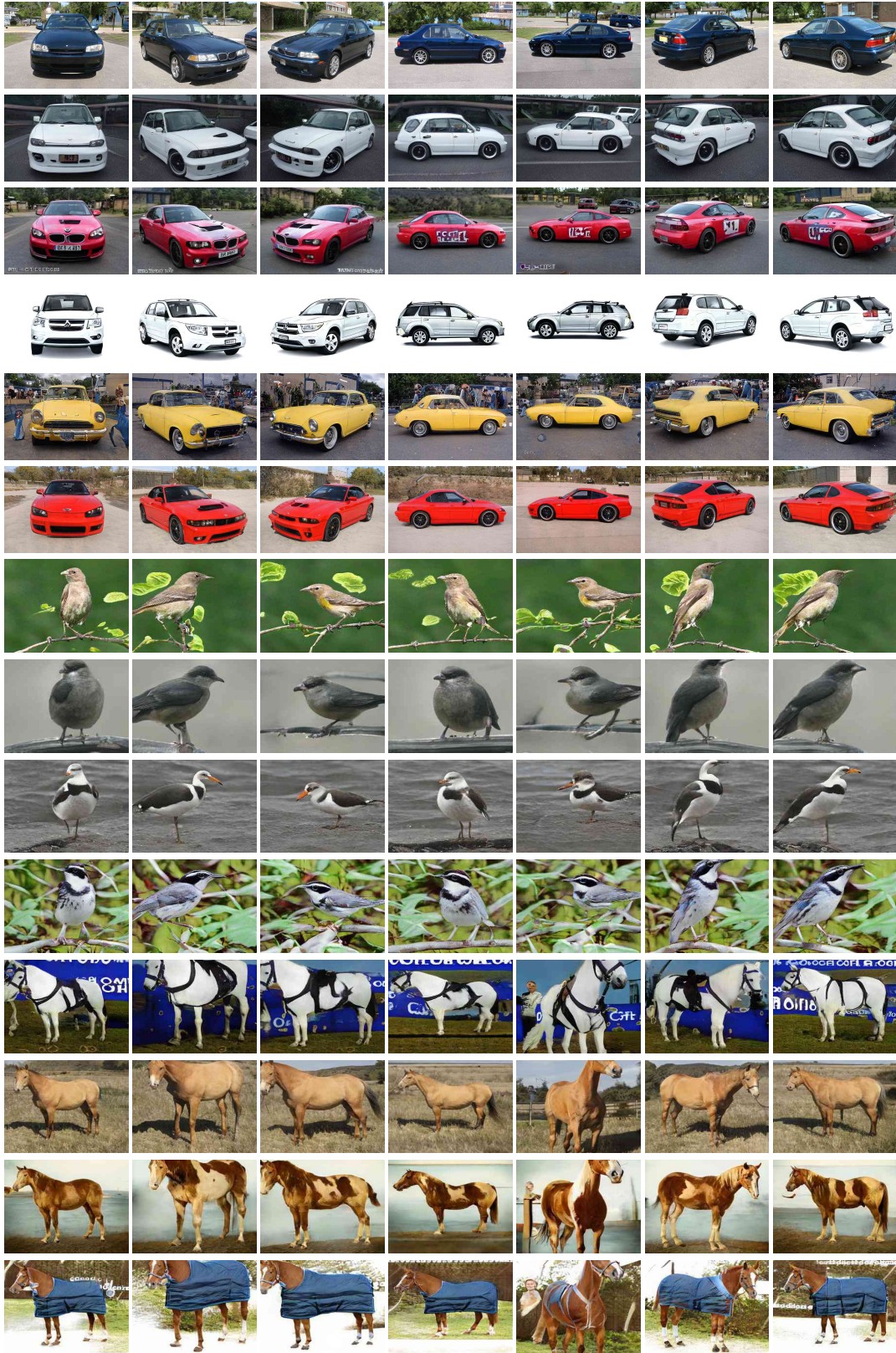

Figure C: **Dataset Overview:** We synthesize multi-view datasets for three classes: *car*, *horse*, and *bird*. Our datasets contain objects with various shapes, textures and viewpoints. Notice the consistency of pose of object in each column (for each class). Challenges include the fact that for all of these objects StyleGAN has not learned to synthesize views that overlook the object from above due to the photographer bias in the original dataset that StyleGAN was trained on.

## D   CAMERA INITIALIZATION

Inverse graphics tasks require camera pose information during training, which is challenging to acquire for real imagery. Pose is generally obtained by annotating keypoints for each object and running structure-from-motion (SFM) techniques (Welinder et al., 2010; Xiang et al., 2014) to compute camera parameters. However, keypoint annotation is quite time consuming – requiring roughly 3-5minutes per object which we verify in practice using the LabelMe interface (Torralba et al., 2010). In our work, we utilize StyleGAN to significantly reduce annotation effort since samples with the same $w_v^*$ share the same viewpoint. Therefore, we only need to assign a few selected $w_v^*$ into camera poses. In particular, we assign poses into several bins which we show is sufficient for training inverse graphics networks where, along with the network parameters, cameras get jointly optimized during training using these bins as initialization.

Specifically, we assign poses into 39, 22 and 8 bins for the car, horse and bird classes, respectively. This allows us to annotate all the views (and thus the entire dataset) in only *1 minute*. We do acknowledge additional time in selecting good views out of several candidates.

We annotate each view with a rough absolute camera pose (which we further optimize during training). To be specific, we first select 12 azimuth angles: [0°, 30°, 60°, 90°, 120°, 150°, 180°, 210°, 240°, 270°, 300°, 330°]. Given a StyleGAN viewpoint, we manually classify which azimuth angle it is close to and assign it to the corresponding label with fixed elevation (0°) and camera distance.

To demonstrate the effectiveness of our camera initialization, we make a comparison with another inverse graphics network trained with a more *accurate* camera initialization. Such an initialization is done by manually annotating object keypoints in each of the selected views ($w_v^*$) of a single car example, which takes about 3-4 hours (around 200 minutes, 39 views). Note that this is still a significantly lower annotation effort compared to 200-350 hours required to annotate keypoints for every single object in the Pascal3D dataset. We then compute the camera parameters using SfM. We refer to the two inverse graphics networks trained with different camera initializations as *view*-model and *keypoint*-model, respectively.

We visualize our two different annotation types in Fig D. We show annotated bins in the top. We annotated keypoints for the (synthesized) car example in the first image row based on which we compute the accurate viewpoint using SfM. To showcase how well aligned the objects are for the same viewpoint code, we visualize the annotated keypoints on all other synthesized car examples. Note that we do not assume that these keypoints are accurate for these cars (only the implied viewpoint).

We quantitatively evaluate two initialization methods in Table. D. We first compare the annotation and training times. While it takes the same amount of time to train, *view*-model saves on annotation time. The performance of *view*-model and *keypoint*-model are comparable with almost the same 2D IOU re-projection score on the StyleGAN test set. Moreover, during training the two camera systems converge to the same position. We evaluate this by converting all the views into quaternions and compare the difference between the rotation axes and rotation angles. Among all views, the average difference of the rotation axes is only 1.43° and the rotation angle is 0.42°. The maximum difference of the rotation axes is only 2.95° and the rotation angle is 1.11°.

We further qualitatively compare the two methods in Fig. E, showing that they perform very similarly. Both, qualitative and quantitative comparisons, demonstrated that *view*-camera initialization is sufficient for training accurate inverse graphics networks and no additional annotation is required. This demonstrates a scaleable way for creating multi-view datasets with StyleGAN, with roughy a minute of annotation time per class.

## E   3D INFERENCE

We here present additional 3D prediction results and compare our model, which is trained on our StyleGAN generated dataset (StyleGAN-model), with the one trained on the Pascal 3D dataset (Xiang et al., 2014) (PASCAL-model). We qualitatively compare two models on the Pascal3D test set in Fig. F and web imagery in Fig. G. Our StyleGAN-model produces better shape and texture predictions in all the testing datasets, which is particularly noticeable when looking at different rendered views of the prediction. We also present additional 3D prediction results on horses and birds in Fig. H.

Azimuth=0°  Azimuth=30°  Azimuth=30°  Azimuth=180°  Azimuth=210°  Azimuth=270°

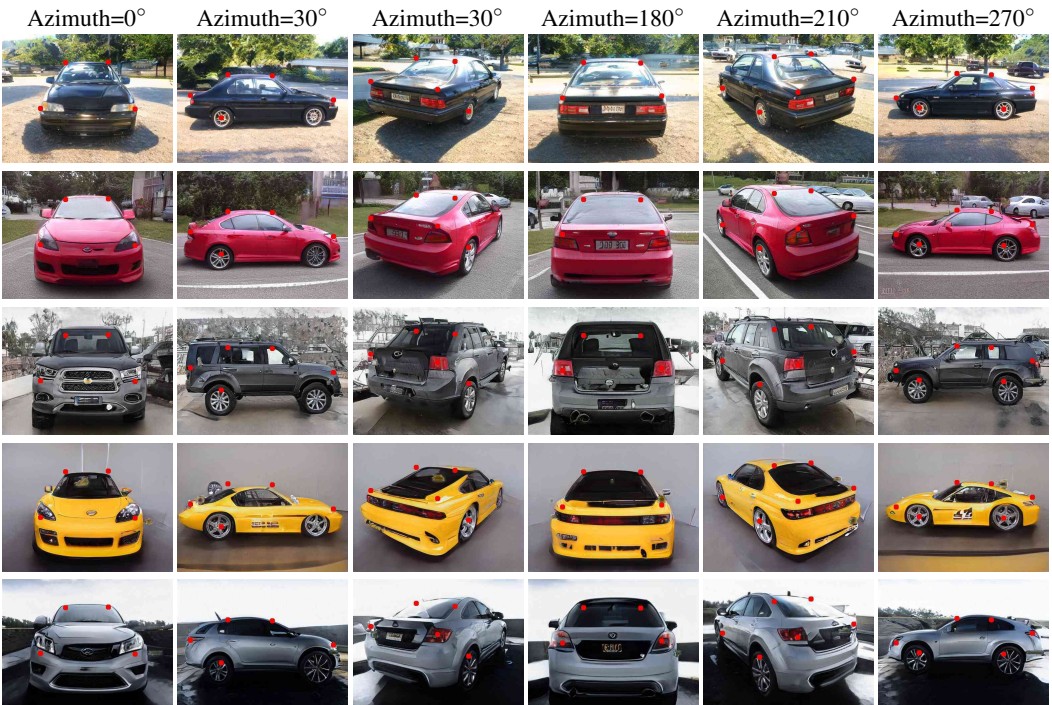

Figure D: We show examples of cars synthesized in chosen viewpoints (columns) along with annotations. Top row shows the pose bin annotation, while the images show the annotated keypoints. We annotated keypoints for the car example in the first image-row based on which we compute the accurate camera parameters using SfM. To showcase how well aligned the objects are for the same viewpoint latent code, we visualize the annotated keypoints on all other synthesized car examples. Note that we do not assume that these keypoints are accurate for these cars (only the implied viewpoint). Annotating pose bins took 1 min for the car class, while keypoint annotation took 3-4 hours, both types of annotations thus being quite efficient. We empirically find that pose bin annotation is sufficient in training accurate inverse graphics networks (when optimizing camera parameters during training in addition to optimizing the network parameters).

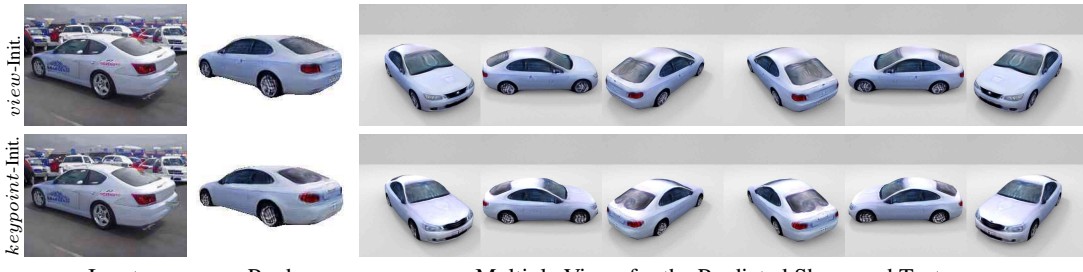

Input          Pred.          Multiple Views for the Predicted Shape and Texture

Figure E: **Comparison of Different Camera Initializations**: The first row shows predictions from *keypoint*-Initialization (cameras computed by running SFM on annotated keypoints) and the second row show results obtained by training with *view*-Initialization (cameras are coarsely annotated into 12 view bins). Notice how close the two predictions are, indicating that coarse viewpoint annotation is sufficient for training accurate inverse graphics networks. Coarse viewpoint annotation can be done in 1 minute.

| Annotation Type | Annotation Time | Training Time | 2D IOU |
|---|---|---|---|
| *keypoint* | 3-4h | 60h | 0.953 |
| *view* | 1min | 60h | 0.952 |

(a) Time & Performance

| Quaternion | Mean | Max |
|---|---|---|
| $q_{xyz}$ | 1.43° | 2.95° |
| $q_w$ | 0.42° | 1.11° |

(b) Camera Difference after Training

Table A: **Comparison of Different Camera Initializations**: First table shows annotation time required for the StyleGAN dataset, and training times of the *view*-model and *keypoint*-model on the dataset with respective annotations (binned viewpoints or cameras computed with SFM from annotated keypoints). The *view*-model requires significantly less annotation time, and its final performance is comparable to the *keypoint*-model. Second table shows the difference of the camera parameters after training both methods (which optimize cameras during training). They converge to very similar camera positions. This shows that coarse view annotation along with camera optimization during training is sufficient in training high accuracy inverse graphics networks.

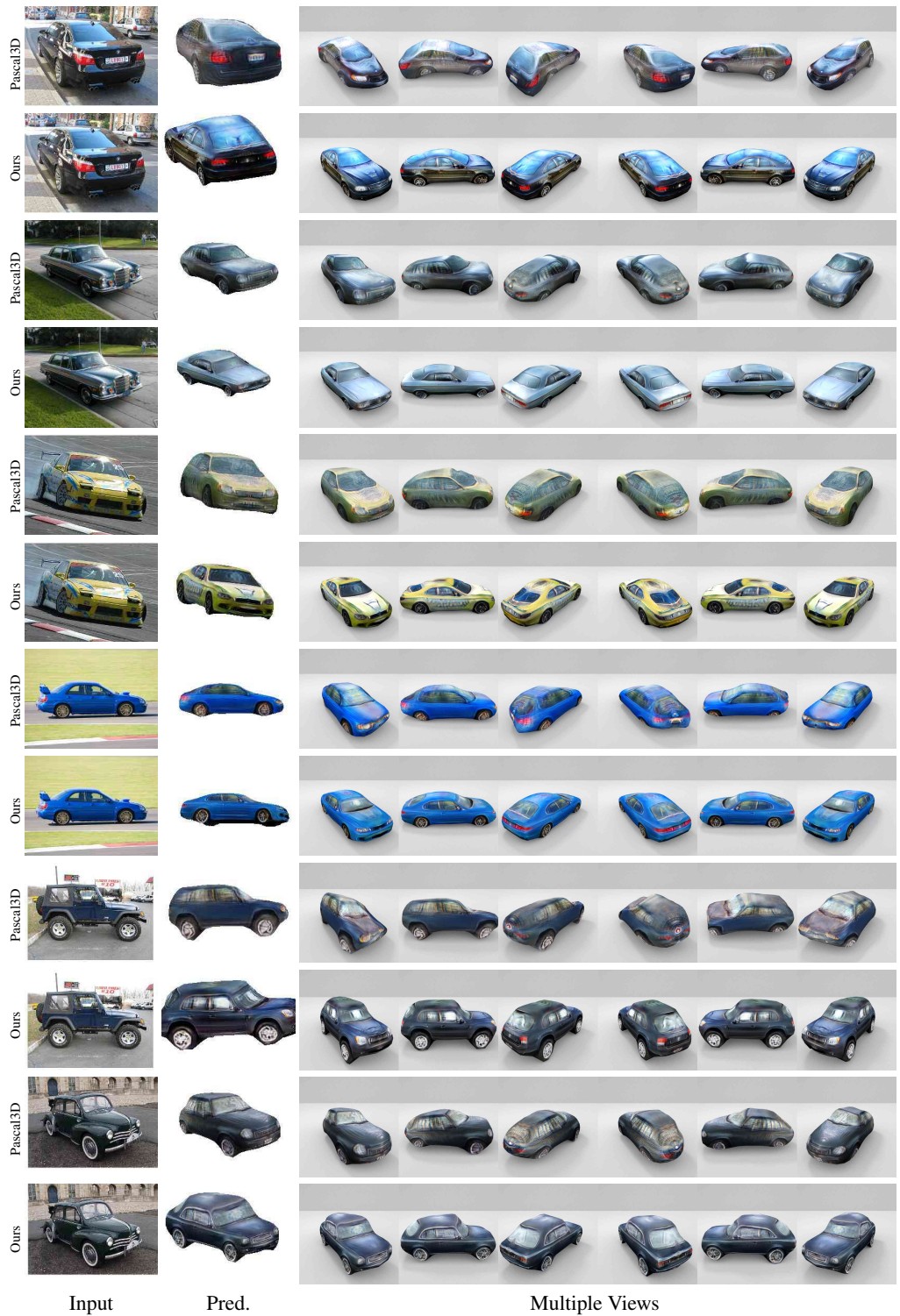

Figure F: **Comparison on PASCAL3D imagery:** We compare PASCAL-model with StyleGAN-model on PASCAL3D test set. While the predictions from both models are visually good in the corresponding image view, the prediction from StyleGAN-model have much better shapes and textures as observed in other views.

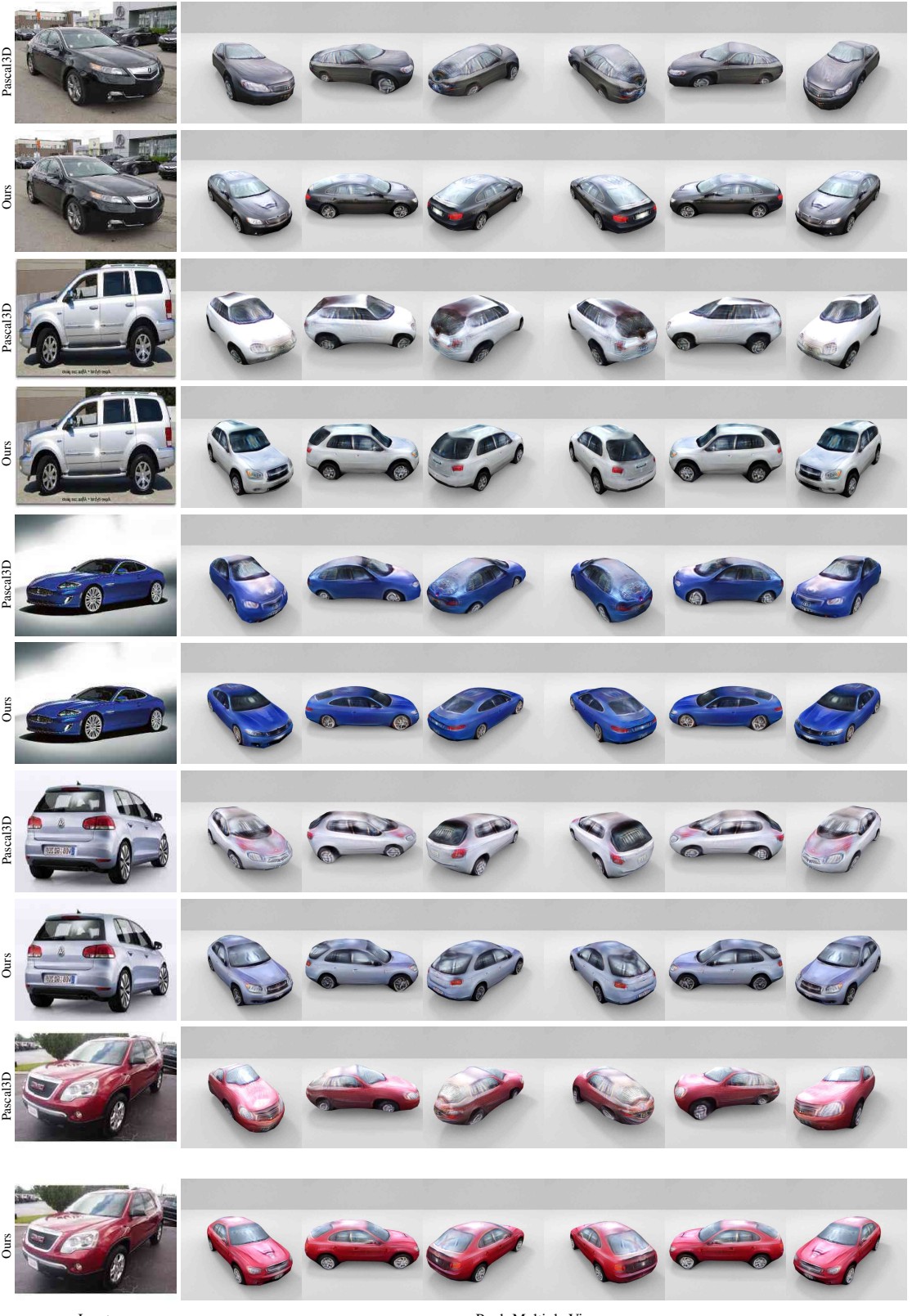

Input                                    Pred. Multiple Views

Figure G: **Comparison on Images from the Web:** We compare the PASCAL-model with our StyleGAN-model on images downloaded from the web. While the predictions from both models are visually good in the corresponding image view, the prediction from StyleGAN-model have much better shapes and textures as observed in other views.

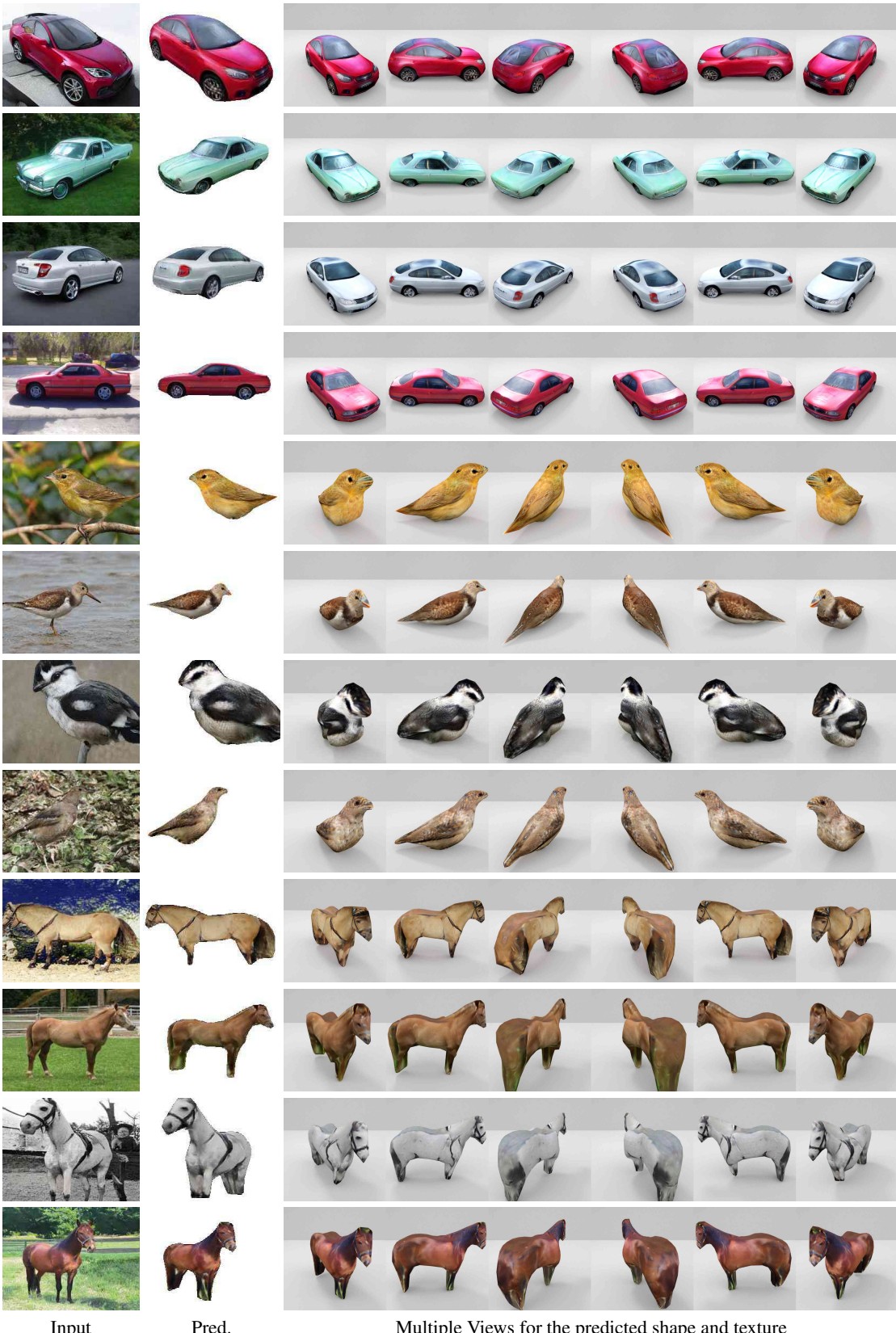

Input        Pred.           Multiple Views for the predicted shape and texture

Figure H: **3D Reconstruction Results for Car, Horse and Bird Classes**: We show car, horse and bird examples tested on the images from the StyleGAN dataset test sets. Notice that the model struggles a little in reconstructing the top of the back of the horse, since such views are lacking in training.

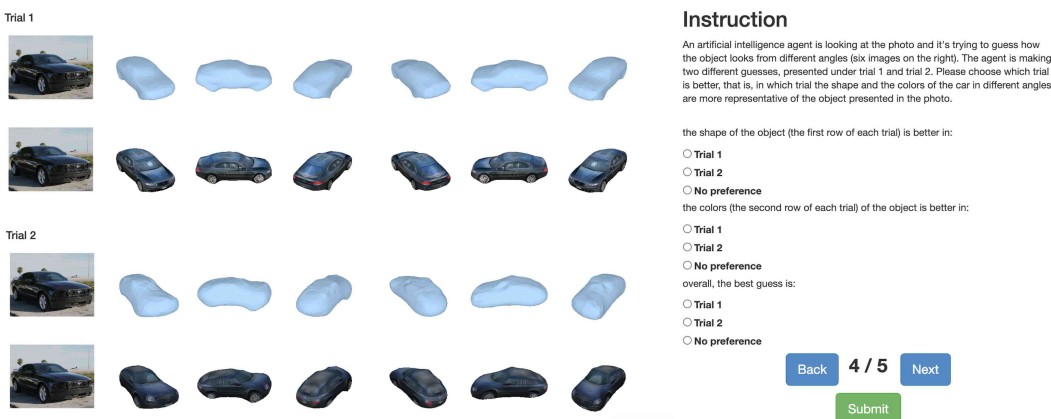

Figure I: **User Study Interface (AMT)**: Predictions are rendered in 6 views and we ask users to choose the result with a more realistic shape and texture that is relevant to the input object. We compare both the baseline (trained on Pascal3D dataset) and ours (trained on StyleGAN dataset). We randomize their order in each HIT.

## F  USER STUDY

We provide user study details in this section. We implement our user interface, visualized in in Fig. I, on Amazon Mechanical Turk. We show the input image and predictions rendered in 6 views such that users can better judge the quality of 3D reconstruction. We show results for both, our inverse graphics network (trained on the StyleGAN dataset) and the one trained on the Pascal3D dataset. We show shape reconstruction and textured models separately, such that users can judge the quality of both, shape and texture, more easily. We randomize the order of ours vs baseline in each HIT to avoid any bias. We ask users to choose results that produce more realistic and representative shape, texture and overall quality with respect to the input image. We separate judgement of quality into these three categories to disentangle effects of 3D reconstruction from texture prediction. We also provide "no preference" options in case of ties. Our instructions emphasize that more "representative" results of the input should be selected, to avoid users being biased by good looking predictions that are not consistent with the input (e.g., such as in the case of overfit networks).

We evaluate the two networks on all 220 images from the Pascal3D test set (which are "in-domain" for the Pascal3D-trained network). For each image we ask three users to perform evaluation, which results in 660 votes in total. We report the average of all votes as our final metric. We further report annotator agreement analysis in Table B. For shape, texture, and overall evaluation, there are 88.2%, 89.2%, and 87.2% cases where at least two out of three users choose the same option.

|  | Overall | Shape | Texture |
|---|---|---|---|
| Ours | **57.5%** | **61.6%** | **56.3%** |
| Pascal3D-model | 25.9% | 26.4% | 32.8% |
| No Preference | 16.6% | 11.9% | 10.8% |

(a) 3D Quality Study

|  | Overall | Shape | Texture |
|---|---|---|---|
| All Agree | **26.1%** | **29.6%** | **27.1%** |
| Two Agree | 61.1% | 58.6% | 62.1% |
| No Agreement | 12.8% | 11.8% | 10.8% |

(b) Annotator Agreement

Table B: User study results: **(a):** Quality of 3D estimation (shape, texture and overall). **(b):** Annotators agreement analysis. "No agreement" stands for the case where all three annotators choose different options.

## G  STYLEGAN-R DISENTANGLEMENT

Given an input image, we infer 3D properties of an object (shape, texture, background) using our inverse graphics network, but can also map these properties back to the latent code and use our StyleGAN-R to synthesize a new image. We show the results in Fig. J. Similar to Fig. 9 in the main paper, we show DIB-R-rendered predictions and neural rendering StyleGAN-R's predictions, and manipulate their viewpoints in rows (1, 4) and (2, 5). We further show "neural rendering" results from the original StyleGAN in row (3, 6), where we only learn the mapping network but keep the StyleGAN's weights fixed. We find that fine-tuning is necessary and StyleGAN-R produces more consistent shape, texture and background.

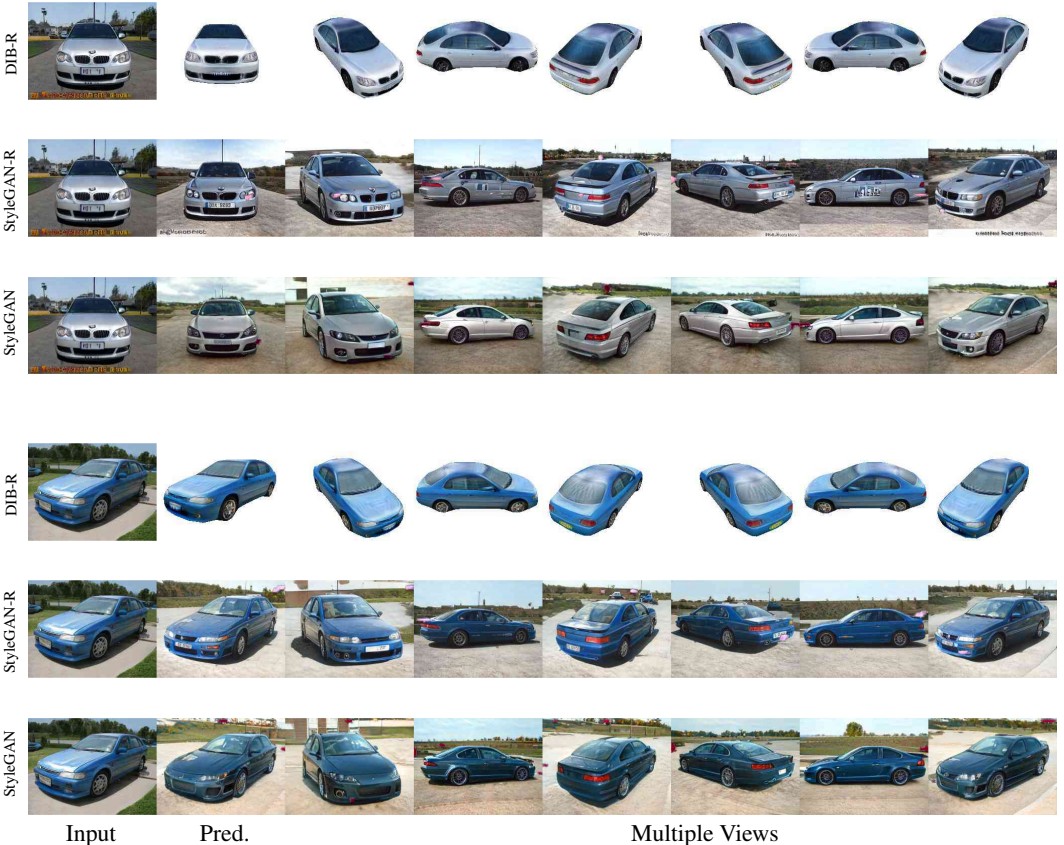

Input          Pred.                                    Multiple Views

Figure J: **Dual Rendering:** Given the input image, we show the DIB-R-rendered predictions in rows (1, 4) and StyleGAN-R's results in rows (2, 5). We further shows the neural rendering results from the original StyleGAN model, where we only learn the mapping network but keep the StyleGAN weights fixed. Clearly, after fine-tuning, StyleGAN-R produces more consistent results.

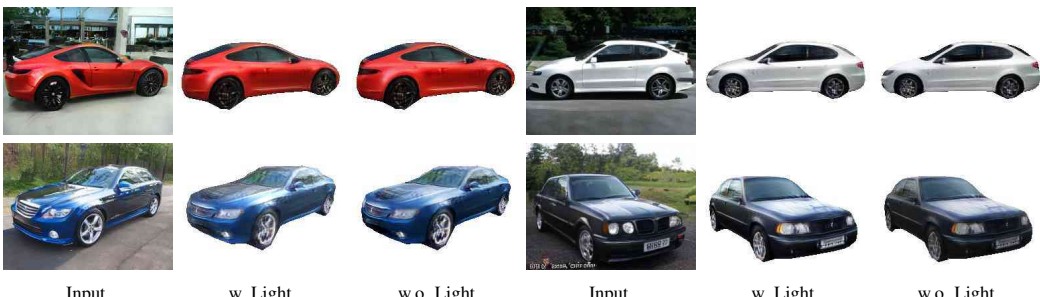

Input          w. Light          w.o. Light          Input          w. Light          w.o. Light

Figure K: **Light Prediction:** Given the input image, we show rendering (using the OpenGL renderer used in DIB-R) results with light (columns 2, 5) and results with just textures (columns 3, 6). We find that the two results are quite similar, which indicates that we did not learn a good predictor for lighting. Moreover, we find that higher order lighting, such as reflection, high-specular light are merged into texture, as shown in the second row. We aim to resolve this limitation in future work.

**Real Image Editing:** We show additional real-image editing examples in Fig. L. With our StyleGAN-R, we can easily change the car's size, azimuth and elevation and synthesize a new image while preserving the shape and texture of the car with a consistent background.

## H  ABLATION STUDIES

We find that the multi-view consistency and perceptual losses play an import role in training, as shown in Fig. P. Multi-view consistency loss helps in training a more accurate inverse graphics network in terms of shape, while the perceptual loss helps to keep texture more realistic.

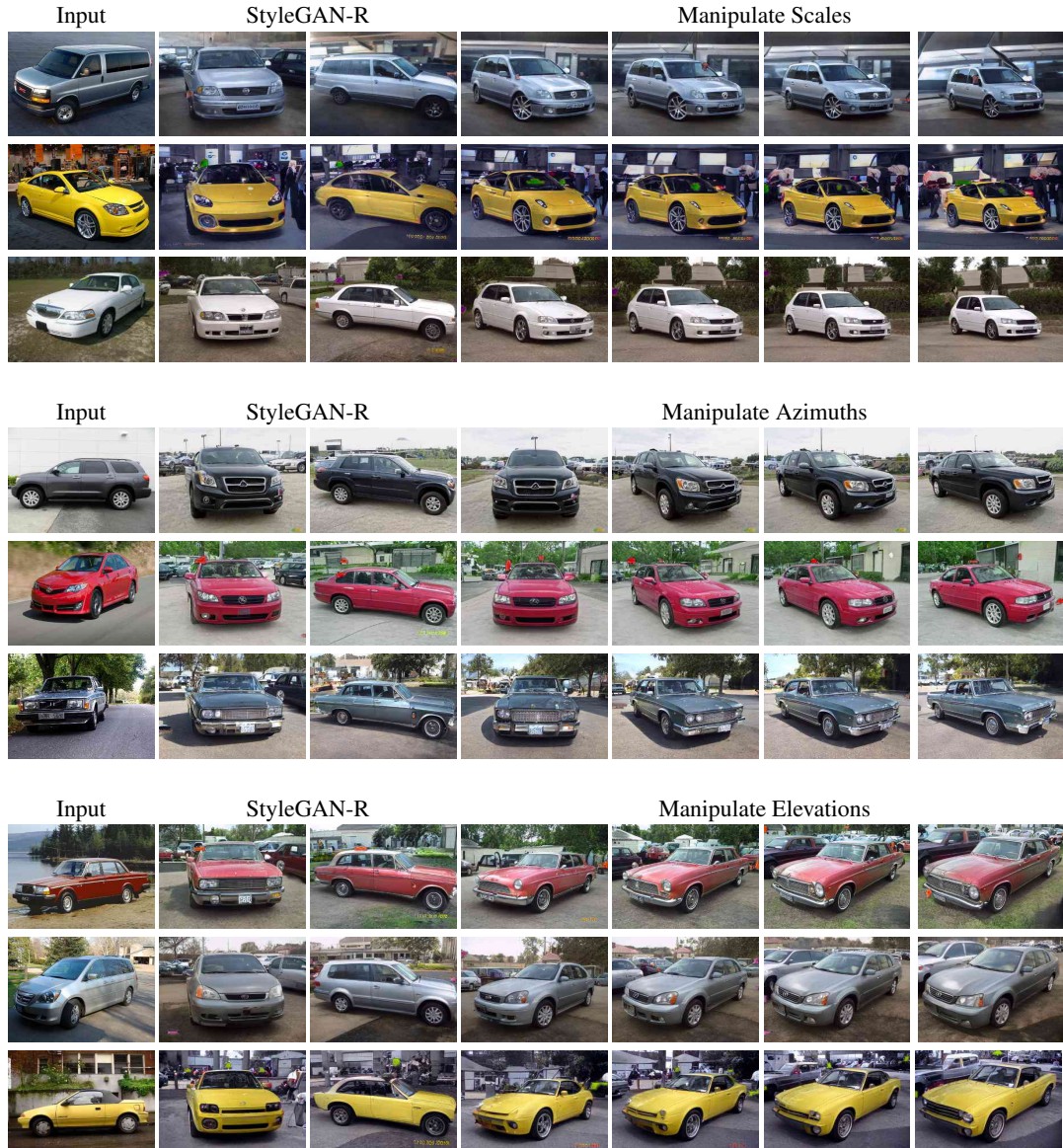

Figure L: **Real Image Editing.** Given an input image (column 1), we use our inverse graphics network to predict the 3D properties and apply StyleGAN-R to re-render these (column 2, 3). We manipulate the car size/scale (row 1-3), azimuth (row 4-6) and elevation (Row 7-9).

## I    STYLEGAN MANIPULATION

We show that our method for manipulating StyleGAN is generalizable and can be generalized to other class, as illustrated in the StyleGAN-R manipulation results for the bird in Fig M and Fig N.

## J    FAILURE CASES

We find that our inverse graphics network fails on out-of-distribution images/shapes, as shown in Fig. O. For example, the reconstruction results for Batmobile and Flinstone cars are not representative of the input cars. We anticipate that this issue can be addressed by augmenting the dataset on which StyleGAN is trained with more diverse objects. Part of the issue is also caused by GANs not capturing the tails of the distribution well, which is an active area of research.

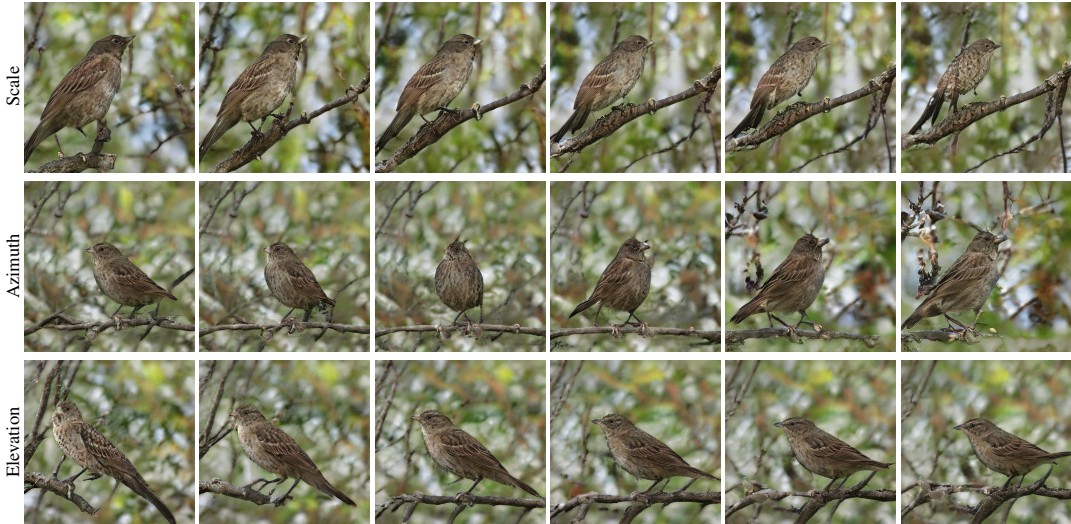

Figure M: **Bird Camera Controller:** We manipulate azimuth, scale, elevation parameters with StyleGAN-R to synthesize images in new viewpoints while keeping content code fixed.

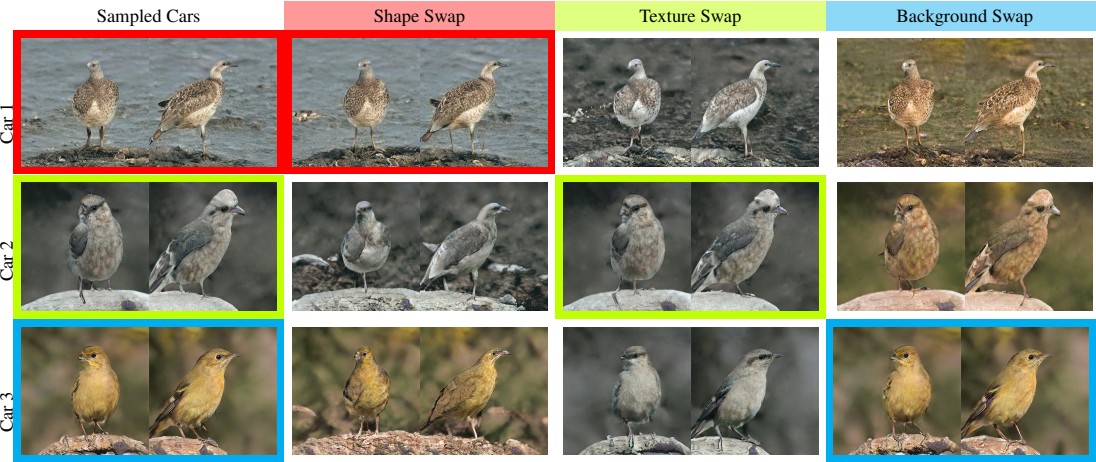

Figure N: **Bird 3D Manipulation:** We sample 3 birds in column 1. We replace the shape of all birds with the shape of Bird 1 (red box) in 2nd column. We transfer texture of Bird 2 (green box) to other birds (3rd col). In last column, we paste background of Bird 3 (cyan box) to the other birds. Examples indicated with boxes are unchanged.

## K    LIMITATIONS

Our simple spherical harmonics model fails to separate light from textures. We show several examples in Fig.K. We leave this issue for future work.

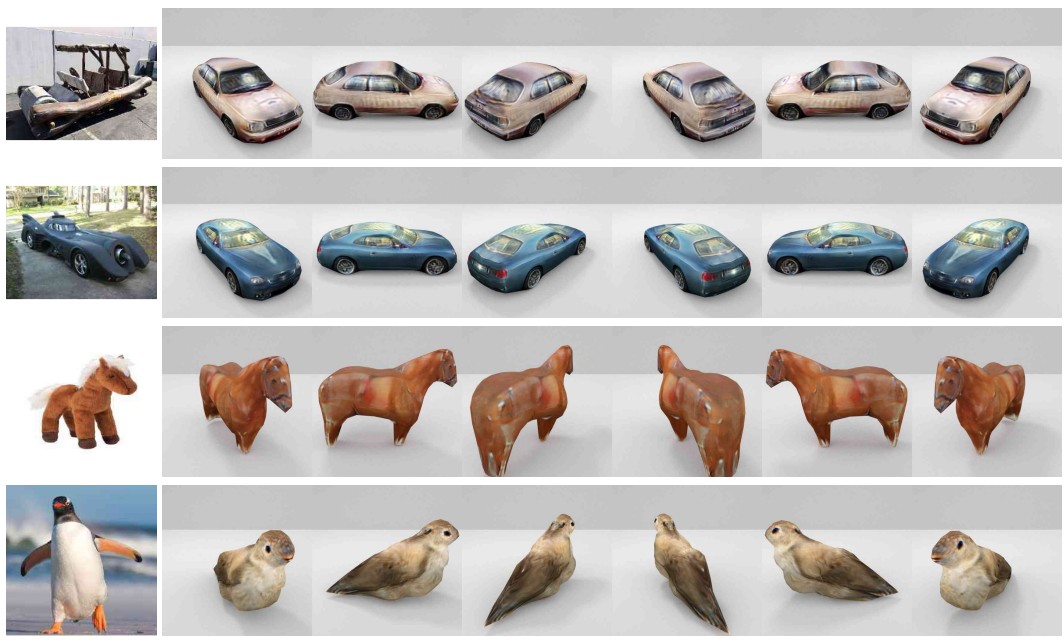

Input            Multiple Views for the predicted shape and texture

Figure O: **3D Reconstruction Failure Cases**: We show examples of failure cases for car, bird and horse. Our method tends to fail to produce relevant shapes for objects with out-of-distribution shapes (or textures).

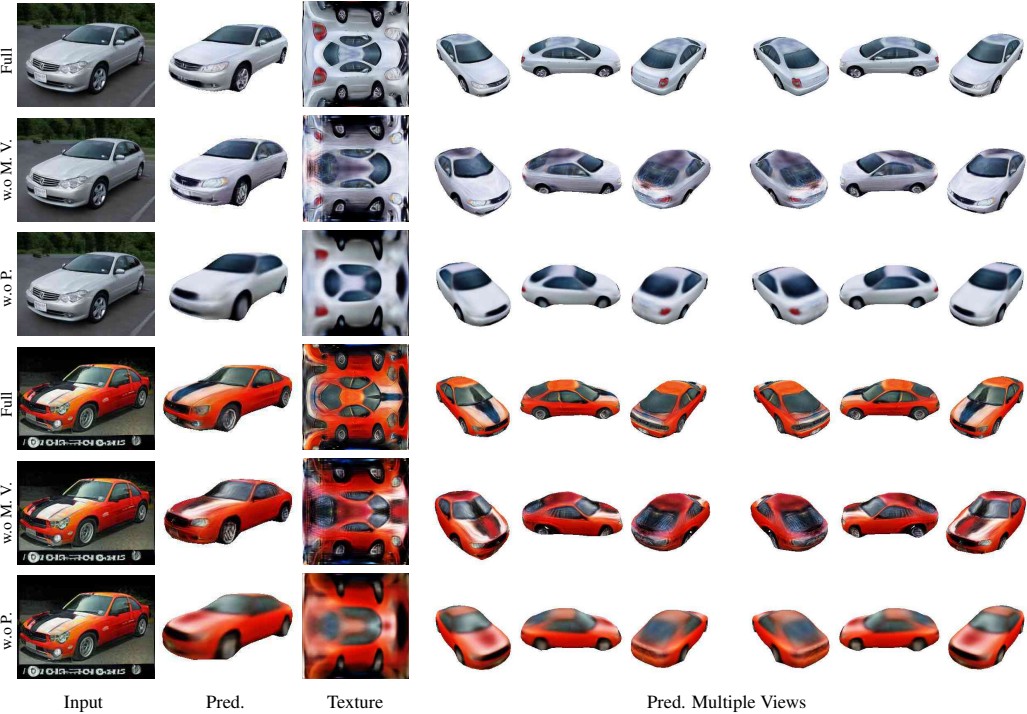

Input     Pred.     Texture                 Pred. Multiple Views

Figure P: **Ablation Study:** We ablate the use of multi-view consistency and perceptual losses by showing results of 3D predictions. Clearly, the texture becomes worse in the invisible part if we remove the multi-view consistency loss (rows 2, 5, denoted by "w.o M. V.", which denotes that no multi-view consistency was used during training), showcasing the importance of our StyleGAN-multivew dataset. Moreover, the textures become quite smooth and lose details if we do not use the perceptual loss (rows 3, 6, noted by "w.o P.", which denotes that no perceptual loss was used during training).

