# OpenReview forum: "Image GANs meet Differentiable Rendering for Inverse Graphics and Interpretable 3D Neural Rendering"
_ICLR.cc/2021/Conference — ICLR 2021 Oral_

### Official Review · AnonReviewer2 · 2020-10-26
**review before discussion: nice stories, unsatisfactory solutions, and nice proof of concept results**

**Rating:** 6
**Confidence:** 3

**Review:**

This paper proposes to use StyleGAN images to train a differential renderer, then in turn to use this differential renderer to train a more controllable GAN (dubbed StyleGAN-R) which allows to perform independent manipulation of shape, background, and texture. There are many different things in this paper, and I am struggling a bit to decide what's the main contribution, here are the ones that seem the main one to me (note I am neither a GAN nor a neural renderer expert):

1. training a neural renderer with GAN images: to the best of my knowledge, that has never been demonstrated and is nice in itself. However, the way this is achieved is not completely satisfying: the entire idea is based on the empirical observation that the first part  of styleGAN latent codes correspond to viewpoint; the training requires manual selection of "good" latent codes and annotation of the viewpoints (once, this is fast, but remains very unsatisfying and strongly limits the number of viewpoints); further image filtering is required using a trained MaskRCNN and masks are necessary for training. Additionally, while this is nice to see, there seem to be very little/no technical contribution in this part which is mainly engineering and empirical results

2. Another way to see the first contribution is to say that the paper demonstrate that using GAN generated images allows to improve over using datasets of manually annotated images. This has been an important hope as an application of GAN, and the beginning of the results section (4.1) seems to emphasize this aspect: less annotation time and higher quality than training a differential renderer on Pascal3D. Again, while this would make a very nice story, this is not related to any technical contribution. Additionally, it's hard to really trust the results on this, since the training of the networks are complicated and the results are hard to evaluate, it might be that the hyperparameters are simply better adapted to training with the StyleGAN training set than with Pascal3D (the presented AMT experiment is good for evaluation, but it is too expensive to do to use it to choose training hyperparameters)

3. StyleGAN-R for decorelated image manipulations. This is again very appealing, but done in an unsatisfying way and with limited results (only on cars). There are many tricks involved in making this work, from the choice of 144 (!?) dimension among the 2048 for the latent vector to the complicated stage-wise training. The results of this step look good, but are only shown for cars, since reconstruction was also done for horses and birds, I can only assume that this doesn't work for other categories.

To summarize, I think there are several cool stories in this paper and some appealing results. While the methods remain unsatisfying, the paper shows interesting proof of concepts, I would thus tend to accept it, but I could be convinced otherwise especially if other reviewers who know the field better than I do point to other papers demonstrating similar points.

I list bellow more specific issues/requests:

a. whether it works or not, I want to see the equivalent of figures 8-9-10 with other categories. This should also be discussed in limitations (or the paper should demonstrate it works). I am annoyed that this is not discussed more visibly in the paper and kind of hidden. No clear answer on this point would lead me to recommend rejection (but I could still recommend accept even if the method doesn't work very well on other categories)

b. I would like to see a comparison between StyleGAN (using the annotated viewpoints) and styleGAN-R for camera controler (figure 8, but using the code predicted by the approach before styleGAN finetuning - not the optimized code as in fig 7), that seems the natural baseline

c. I don't agree at all with the claim in 4.1 that the approach works with articulated classes: horses are indeed articulated, but looking at the horse dataset used (figure C in appendix) reveals that the selected images from styleGAN do *not* present any articulation (on the contrary to most real horse datasets). Thus the approach works simply because styleGAN do not present the same diversity as natural images, and would not work for actual articulated horses, this is more a limitation that a strength.

d. fig 7 comparison with optimization: to me the natural approach/baseline would be to use a much simpler CNN to predict the latent code (and potentially do some local optimization afterward). This type of learning for optimization approach often regularize the problem well, and lead to good results. This would also be more similar in spirit to the proposed approach.

---

> ### Author Response · Authors · 2020-11-24
> **Thanks for the detailed comments (1)**
>
> We thank the reviewer for detailed comments. We address the concerns below:
>
> 1. Employing GANs to synthesize multi-view data to train the inverse graphics networks is a novel idea which we show to perform really well in practice, as also recognized by reviewers R1 and R2. While we adopt prior work on differentiable rendering and build our method on top of empirical observations,  “the specific combination of different components is novel and quite creative” (R2).
>
>  Overall, it seems that the reviewer tends to appreciate work that proposes new mathematical formulations, but is less inclined to see value in creative ideas with stronger empirical aspects. We respectfully disagree with this point of view, and believe that our work will lead to wide adoption in the community of utilizing GANs for inverse graphics tasks.
>
>  Finding hyperparameters (such as the dimensions to control the viewpoint) is typically done for all work with an empirical aspect. We use Mask-RCNN since we need masks to indicate the object of interest, as is required in most of the existing work on differentiable rendering (including DIB-R). Mask-RCNN further helps us in filtering irrelevant or noisy images as StyleGAN also sometimes fails in producing realistic objects. In the future, one may consider doing the entire method in an even more elegant way, but this will also necessitate training even more powerful GANs, and relieving the reliance of differentiable renderers on segmentation masks.
>
>
> 2. We show strong advantages of our method over inverse graphics networks that are trained on existing (labeled!) datasets such as Pascal3D, in terms of both the dataset size and annotation time, as well as the final 3D reconstruction performance. It is also worth mentioning that we are able to generate multi-view images, which is crucial in the task of single image 3D reconstruction. Our method allows us to create multi-view datasets out of unorganized online photo collections, and we believe such datasets (along with the method to create them) will greatly benefit the community. Please see 1) for our belief in what is a valuable scientific contribution.
>
>  Regarding hyperparameters: We use the exact hyperparameters from DIB-R that were used in their paper when applying their method to Pascal3D, which is also our baseline. The comparison is fair, if not favourable to the baseline. We plan to release code for reproducibility.
>
>
> 3. We thank the reviewer for the suggestion of showing StyleGAN-R on other classes. Please see a revised manuscript for these results.

---

> > ### Author Response · Authors · 2020-11-24
> > **Thanks for the detailed comments (2)**
> >
> > Experiments:
> >
> > 1. StyleGAN-R manipulation on other classes: Our method is general and it can be applied to other classes as well. We further validate our method on the bird dataset and get similar results -- please see a revised manuscript (Fig. P,Q in appendix) for these results. Due to a logistics issue we cannot show results on horse in time for the rebuttal, which we will add to the final manuscript.
> >
> >
> > 2. StyleGAN v.s. StyleGAN-R: We have shown such examples in Fig.6, where we refer to dual renders. Given StyleGAN images (Col.1,4), we predict mesh and texture, and render them with the graphics renderer (Col. 2,5), and finally map them back to latent codes and adopt StyleGAN-R to render them (Col. 3.6).
> >
> >  The main difference in camera manipulation of StyleGAN-R is that StyleGAN-R accepts explicit camera parameters as input, and thus viewpoints are more easily controllable. Original StyleGAN does offer control over viewpoints, but in an implicit way, by sampling viewpoint codes and labeling the camera. We clarified this in the revised manuscript in Fig.8, and thank the reviewer for bringing this point up.
> >
> >  The reviewer is right in that we can sample the exact same cameras as those corresponding to the “viewpoint” latent codes we discovered in StyleGAN, to verify that the synthesized views look equally good. These results can be found in Fig. R in appendix.
> >
> >
> > 3. There seems to be a misunderstanding here. We do NOT claim that our approach handles articulated objects anywhere in our paper. In fact, we currently only handle articulated objects such as birds and horses by treating them as rigid objects that always adopt the same articulation. We specifically choose an articulation, and try to keep it fixed across the different viewpoints. In fact, articulation is not perfectly disentangled in StyleGAN, and thus there are slight variations in articulation across the different viewpoints. We discussed this point in the supplementary, section C: “Note that for articulated objects such as horse and bird, StyleGAN does not perfectly preserve the articulation in different viewpoints, which leads to challenges in training high accuracy models using multi-view consistency loss. We leave further investigation in articulated objects to future work.”
> > We have further emphasized limitations with respect to object articulations in the revised manuscript. Handling articulated objects is a fruitful direction for future work.
> >
> >
> > 4. We followed Image2StyleGAN: How to Embed Images Into the StyleGAN Latent Space? to create Fig 7. We agree that using an encoder to predict the latent code can potentially lead to better results for StyleGAN and we add this experiment in Fig. S. StyleGAN’s synthesis in this case is improved significantly. The texture is also more faithful than for StyleGAN-R, which means that further improvements in texture map prediction (which are input to StyleGAN-R) for the inverse graphics network are possible. We emphasize that StyleGAN is not a baseline here, as this is really the meat of StyleGAN-R -- our main point is that we turn StyleGAN into a neural renderer that can accept explicit geometry texture and background and ``render” a photo-realistic image (just as a graphics renderer does).

---

### Official Review · AnonReviewer1 · 2020-10-28

**Rating:** 8
**Confidence:** 3

**Review:**


This paper proposes to couple a GAN, an inverse graphics network, and a differentiable renderer. The authors base their work on StyleGAN, and use the observation that a specific part of the latent code corresponds to camera view-point to rapidly annotate a large amount of synthetic images with approximate camera pose. They then use these images and rough annotations to train the inverse graphics network to provide 3D and texture data. The differentiable renderer is used to synthesize 2D images from 3D, which can be compared to the input for consistency. In a second step, the authors use the inferred 3D data to disentangle the latent space of StyleGAN to allow to use it as a controllable renderer.

--- Strengths ---

The presented results look great. The inverse graphics network, seems to get both shape and textures mostly right. This is especially remarkable, given the small amount of supervision that was used. The controllable StyleGAN provides very plausible results. Most importantly, results  on real images are shown, and indicate the the presented approach does make progress in brining inverse graphics networks to real images.

While the individual parts of this work are taken from other prior works, the specific combination of different components is novel and quite creative. Using a GAN to train an inverse graphics network is to the best of my knowledge novel. So is using an inverse graphics network to turn a GAN into a controllable renderer. There are also various key-insights that make the approach work, such as the that StyleGAN is partially  disentangled with respect to view-point, or the use of multi-view consistency when training the inverse graphics network.

--- Weaknesses ---

The paper is overall well written, but leaves out some technical descriptions which make it not self-contained and hard to reproduce. Specifically, I'd like to ask the authors to elaborate more on the individual loss terms in Equation (1). Especially, the 3D-related losses $L_{lap}$ and $L_{mov}$ are unclear.

While many results are shown in the supplement, I would have loved to see a video that shows more results. For example, rotating the obtained 3D models, or showing interpolations of various factors for the controllable renderer. Even if the results are not perfect, a video would help to judge the overall quality of the results.

--- Summary ---

This paper shows an interesting pipeline with good results. It presents some non-trivial observations and effectively leverages them into a complete system that looks quite impressive.

Typos:

Page 3, last sentence: "conten"

--- Post rebuttal ---
After reading the other reviewers comments and the rebuttal, I'm keeping my initial score.

---

> ### Author Response · Authors · 2020-11-24
> **Thanks for the detailed comments**
>
> We thank the reviewer for detailed comments. We will incorporate the suggested modifications, and address each individual concern below:
>
> 1. During training, we learn both a category template shape (independent of an input image) and the vertex offsets (dependent on input image). At test time, the template is fixed during inference, and we reason about vertex offsets deviating from this template shape. The vertices of the predicted shape are obtained by summing the template shape vertices and predicted offsets. The intuition behind using $L_{lap}$ is to regularize the offsets of neighboring vertices, e.g. neighbouring vertices should have similar offsets to avoid large local deformations. On the other hand, $L_{mov}$ is used to encourage the offsets to be small such that the predicted shape will not differ too much in the overall geometry from the template shape. We use a weighting to balance the different loss terms. We will add more explanation in the next revision.
>
> 2. We strongly agree with the idea of making a video to show the animation of 3D models and StyleGAN-R interpolation. We have prepared such a video to demonstrate the 3D reconstruction models in Fig.4 and camera changes in Fig.8.

---

### Official Review · AnonReviewer3 · 2020-10-31
**Inverse graphics work with impressive results**

**Rating:** 8
**Confidence:** 4

**Review:**

The authors provided a framework for inverse graphics, i.e., infer 3D mesh, light, and texture from a 2D image. They first use StyleGAN to generate realistic multi-view images, and then trained their model with a differentiable graphics renderer.

Pros
- Training on data generated by StyleGAN is novel. It is easy to control the view of rendered images, but they usually do not look real. Real images look real, but we usually do not know the viewpoint (most annotated datasets are either small or have limited annotation quality). The authors proposed a method to generate images of the same category of objects with the same viewpoint (Figure 2), which addresses this issue -- the viewpoints are known, and the images look real.
- The experimental part is impressive. 3D reconstructions look realistic, and the authors demonstrated the effectiveness to train on StyleGAN by comparing their model to a neural network trained on PASCAL3D+.
- The way the authors generate data by StyleGAN is also novel. They empirically realized some latent code in StyleGAN controls the camera viewpoint, so that they only need to choose some latent codes that well spanned over the space. It is a wise and elegant way to control the viewpoints of an object in StyleGAN.

In summary, compared to previous methods, generating multi-view images from StyleGAN solves the unrealistic and unknown viewpoint issue, and the results look impressive.

---

> ### Author Response · Authors · 2020-11-24
> **Thanks for the detailed comments**
>
> We thank the reviewer for detailed comments. We are happy that the reviewer appears as excited about the potential of this work as we are. Utilizing GANs to synthesize a multiview dataset opens a new door in training inverse graphics by leveraging widely available photo collections, and we demonstrate the effectiveness of this idea. We will release the source code for full reproducibility.

---

### Decision · Program_Chairs · 2021-01-07
**Final Decision**

**Decision:**

Accept (Oral)

**Comment:**

The paper proposes to bring together a GAN, a differentiable renderer, and an inverse graphics model. This combined model learns 3D-aware image analysis and synthesis with very limited annotation effort (order of minutes). The results look impressive, even compared to training on a labeled dataset annotation of which took several orders of magnitude more time.

The reviewers point out the novelty of the proposed system and the very high quality of the results. On the downside, R2 mentions that the model appears over-engineered and some important experimental results are missing. The authors’ response addresses these concerns quite well.

Overall, this is a really strong work with compelling results, taking an important step towards employing generative models and neural renderers “in the wild”. I think it can make for a good oral.